# Polynomial-Time Computation of Optimal Correlated Equilibria in Two-Player Extensive-Form Games with Public Chance Moves and Beyond

**Gabriele Farina**
Computer Science Department
Carnegie Mellon University
gfarina@cs.cmu.edu

**Tuomas Sandholm**
Computer Science Department, CMU
Strategy Robot, Inc.
Strategic Machine, Inc.
Optimized Markets, Inc.
sandholm@cs.cmu.edu

## Abstract

Unlike normal-form games, where correlated equilibria have been studied for more than 45 years, *extensive-form* correlation is still generally not well understood. Part of the reason for this gap is that the sequential nature of extensive-form games allows for a richness of behaviors and incentives that are not possible in normal-form settings. This richness translates to a significantly different complexity landscape surrounding extensive-form correlated equilibria. As of today, it is known that finding an optimal *extensive-form correlated equilibrium (EFCE)*, *extensive-form coarse correlated equilibrium (EFCCE)*, or *normal-form coarse correlated equilibrium (NFCCE)* in a two-player extensive-form game is computationally tractable when the game does not include chance moves, and intractable when the game involves chance moves. In this paper we significantly refine this complexity threshold by showing that, in two-player games, an optimal correlated equilibrium can be computed in polynomial time, provided that a certain condition is satisfied. We show that the condition holds, for example, when all chance moves are *public*, that is, both players observe all chance moves. This implies that an optimal EFCE, EFCCE and NFCCE can be computed in polynomial time in the game size in two-player games with public chance moves.

## 1 Introduction

A vast body of literature in computational game theory has focused on computing Nash equilibria (NEs) in two-player zero-sum imperfect-information extensive-form games. Success stories from that endeavor include the creation of strong—in some cases superhuman—AIs for several complex games, including two-player limit Texas hold'em [4], two-player no-limit Texas hold'em [5, 6, 22], and multiplayer no-limit Texas hold'em [7]. NE captures strategic interactions in which each player maximizes her own utility. The interaction in NE is assumed to be fully decentralized: no communication between players is possible and the behavior of the players is not coordinated by any external orchestrator in any way. While that assumption is natural in games such as poker, NE is too restrictive in other types of strategic interactions in which partial forms of communication or centralized control are possible [1]. Therefore, there has been growing interest around less restrictive solution concepts than NE.

*Correlated* and *coarse correlated equilibria* are classic families of solution concepts that relax the assumptions of NE to allow forms of coordination of utility-maximizing agents [3, 23]. In correlated and coarse correlated equilibria, a mediator that can recommend behavior—but not enforce it— complements the game. Before the interaction starts, the mediator samples a profile of recommended

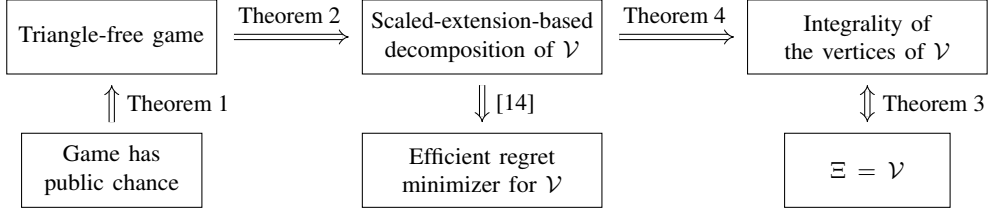

Figure 1: Overview of the connections among this paper's results.

strategies (one for each player) from a publicly known correlated distribution. The mediator reveals the next recommended move (or sequence of moves, depending on the specific solution concept in the family) to each acting player. In correlated equilibrium, each agent must decide whether to commit to following the next recommended move (or sequence of moves) *after* such move or sequence of moves is revealed by the mediator. In coarse correlated equilibrium, each agent must decide whether to commit to following the next recommended move (or sequence of moves) *before* it is revealed by the mediator. If a player chooses not to follow the recommendation, the mediator stops issuing further recommendations to that player. Since the selfish agents are free to not follow the recommendations, it is up to the mediator to come up with a correlated distribution of recommendations such that no agent has incentive to deviate from the recommendations, assuming no other player deviates. Despite the apparent weakness of a mediator that cannot enforce behavior but only suggest it, the maximum social welfare (that is, sum of the players' utilities) that can be induced by these families of solution concepts is greater than the social welfare obtainable by NE. Examples of interactions where a mediator is natural include traffic control and load balancing [1].

These equilibrium concepts have typically been studied in normal-form (that is, matrix) games. The study of correlation in *extensive-form* (that is, tree-form) games is recent, and was pioneered by von Stengel and Forges [26]. Three correlated solution concepts are often used in extensive-form games: *extensive-form correlated equilibrium (EFCE)* [26], *extensive-form coarse correlated equilibrium (EFCCE)* [15], and *normal-form coarse correlated equilibrium (NFCCE)* [23, 8, 9]. Compared to normal-form (that is, one-shot) games, extensive-form correlation poses new and different challenges, especially in settings where the agents retain private information. This is unique to the sequential nature of extensive-form games, where, fundamentally, players can adjust strategically as they make observations about their opponents and the environment [13]. These challenges also translate to some negative complexity results for extensive-form correlation [17, 26]. While a landmark positive complexity result in game theory shows that *one* EFCE, EFCCE, or NFCCE can be found in polynomial time [24, 18, 19], the computation of an *optimal* (that is, one that maximizes or minimizes a given linear objective, such as social welfare) EFCE, EFCCE, or NFCCE is computationally intractable in games with more than two players, as well as two-player games with chance moves, and tractable in two-player games without chance moves [26].

In this paper we significantly refine this complexity threshold by showing that, in two-player games, an optimal correlated equilibrium can be computed in polynomial time, provided that a certain *triangle-freeness* condition—which can be checked in polynomial time—is satisfied. We prove that the condition holds, for example, when all chance moves are *public*, that is, both players observe all chance moves. This includes, for example, games where the chance outcomes amount to public dice rolls or public revelations of cards. Specifically, we show that the set of *correlation plans* $\Xi$ of a triangle-free game coincides with the *von Stengel-Forges polytope* $\mathcal{V}$ of the game—a polytope that only requires a polynomial number of linear "probability-mass-conserving" constraints. Since $\mathcal{V}$ can be represented using a polynomial number of constraints in the input game size, optimizing over this set can be efficiently done by means of, for example, linear programming methods.

In Figure 1 we give an overview of the results in this paper and how they relate to each other. Our main result is that the polytope of correlation plans $\Xi$ coincides with the von Stengel-Forges polytope $\mathcal{V}$ when the game satisfies the *triangle-freeness* condition that we introduce (Definition 3). As we show in Theorem 1, every two-player game with public chance moves (which includes games with no chance moves at all) is triangle-free, but not all triangle-free games have public chance moves. So, our results also apply to some games where chance is not public. The equality $\Xi = \mathcal{V}$ in triangle-free games implies that an optimal EFCE, EFCCE and NFCCE can be computed in polynomial time. This

is because $\mathcal{V}$ has a polynomial (in the game size) description [26] and the computation of an EFCE, EFCCE, NFCCE can be expressed as a linear program [26, 15].

We prove $\Xi = \mathcal{V}$ in several steps. First, we show that in triangle-free games, $\mathcal{V}$ admits a structural decomposition in terms of *scaled extension* operations. This type of decomposition of $\mathcal{V}$ was introduced by Farina et al. [14] as a way of "unrolling" the combinatorial structure of $\mathcal{V}$ to construct an efficient regret minimization algorithm for $\Xi$ in two-player games without chance moves. We extend their construction to handle any triangle-free game. Then, we show a deep connection between the integrality of the vertices of the von Stengel-Forges polytope $\mathcal{V}$ and $\Xi$. Namely, in Theorem 3, we show that $\Xi = \mathcal{V}$ holds if and only if all of $\mathcal{V}$'s vertices have integer $\{0, 1\}$ coordinates. Finally, in Section 4 we prove that $\mathcal{V}$ has integral vertices by leveraging its structural decomposition.

## 2 Preliminaries

**Extensive-form games** *Extensive-form games (EFGs)* are the standard model for games that are played on a game tree. EFGs can capture sequential and simultaneous moves as well as private information. Each node in the EFG belongs to one player. One special player, called the *chance player*, is used to model random stochastic events, such as rolling a die or drawing cards. In this paper, we only consider games that have two players in addition to potentially having a chance player.

Edges leaving from a node represent actions that a player can take at that node. To model private information, the game tree is supplemented with an *information partition*, defined as a partition of nodes into sets called *information sets*. Each node belongs to exactly one information set, and each information set is a nonempty set of tree nodes for the same Player $i$. An information set for Player $i$ denotes a collection of nodes that Player $i$ cannot distinguish among, given what she has observed so far. The symbols $\mathcal{I}_1$ and $\mathcal{I}_2$ denote the information partition of Player 1 and 2, respectively. Let $I_1$ and $I_2$ be information sets for Player 1 and 2, respectively. $I_1$ and $I_2$ are *connected*, denoted $I_1 \rightleftharpoons I_2$, if there exist nodes $u \in I_1$ and $v \in I_2$ such that $u$ is on the path from the root to $v$, or vice versa.

We will only consider *perfect-recall* games, that is, no player forgets what the player knew earlier. As a consequence, all nodes that belong to an information set $I$ share the same set of available actions (otherwise the player acting at those nodes would be able to distinguish among them), which we denote by $A_I$. We define the set of *sequences* of Player $i$ as the set $\Sigma_i := \{(I, a) : I \in \mathcal{I}_i, a \in A_I\} \cup \{\varnothing\}$, where the special element $\varnothing$ is called *empty sequence*. Given an information set $I \in \mathcal{I}_i$, we denote by $\sigma(I)$ the *parent sequence* of $I$, defined as the last pair $(I', a') \in \Sigma_i$ encountered on the path from the root to any node $v \in I$; if no such pair exists we let $\sigma(I) = \varnothing$.

An important concept in extensive-form correlation is *relevance* of sequence pairs. Intuitively, two sequences are relevant if they belong to connected information sets or if either of them is the empty sequence. Formally, a pair of sequences $(\sigma_1, \sigma_2) \in \Sigma_1 \times \Sigma_2$ is *relevant*, denoted $\sigma_1 \bowtie \sigma_2$, if either $\sigma_1$ or $\sigma_2$ or both is the empty sequence, or if $\sigma_1 = (I_1, a_1)$ and $\sigma_2 = (I_2, a_2)$ and $I_1 \rightleftharpoons I_2$. The set of all relevant sequence pairs is denoted $\Sigma_1 \bowtie \Sigma_2$. Given $\sigma_1 \in \Sigma_1$ and $I_2 \in \mathcal{I}_2$, we say that $\sigma_1$ is relevant for $I_2$, and write $\sigma_1 \bowtie I_2$, if $\sigma_1 = \varnothing$ or if $\sigma_1 = (I_1, a_1)$ and $I_1 \rightleftharpoons I_2$ (an analogous statement holds for $I_1 \bowtie \sigma_2$). We say that a sequence $\sigma = (I, a) \in \Sigma_i$ for Player $i$ is *descendent* of another sequence $\sigma' = (I', a') \in \Sigma_i$ for the same player, denoted by $\sigma \succeq \sigma'$, if $\sigma = \sigma'$ or if there is a path from the root of the game to a node $v \in I$ that passes through action $a'$ at some node $v' \in I'$. We use the notation $\tau \succ \tau'$ to mean $\tau \succeq \tau' \wedge \tau \neq \tau'$.

A *reduced-normal-form plan* $\pi_i$ for Player $i$ defines a choice of action for every information set $I \in \mathcal{I}_i$ that is still reachable as a result of the other choices in $\pi$ itself. We denote the set of reduced-normal-form plans of Player $i$ by $\Pi_i$. We denote by $\Pi_i(I)$ the subset of reduced-normal-form plans that prescribe all actions for Player $i$ on the path from the root to information set $I$.

**Polytope of correlation plans ($\Xi$)** A correlated distribution $\mu$ over combinations of plans $\Pi_1 \times \Pi_2$ of the players can be thought of as a point in probability simplex $\Delta^{|\Pi_1 \times \Pi_2|}$. Since the number of plans of each player is exponential in the game tree size, so is that representation of $\mu$. Therefore, von Stengel and Forges [26] introduced a more compact representation of $\mu$, called the *correlation plan representation*. The set of all legal correlation plans is denoted $\Xi$ and called the *polytope of correlation plans*. The set $\Xi$ is a convex polytope in $\mathbb{R}_{\geq 0}^{|\Sigma_1 \bowtie \Sigma_2|}$, so the number of variables is at most quadratic in the game tree size. However, it might still require an exponential number of *constraints*.

An optimal EFCE, EFCCE, or NFCCE is an optimal correlation plan subject to a set of linear *incentive constraints* [26, 13, 15]. These constraints encode the requirement that the set of corrrelated behavior be *incentive compatible* for the player, that is, that no player would be better off not following the recommended behavior than to always follow it. Hence, an optimal EFCE, EFCCE, or NFCCE can be computed as the solution of a linear program. Furthermore, the linear program can be solved in polynomial time if and only if $\Xi$ can be described with a polynomial number of linear constraints. Thus the characterization of the constraints that define $\Xi$ in various classes of games is important.

**The von Stengel-Forges polytope ($\mathcal{V}$)**    The characterization of the constraints that define $\Xi$ was initiated by von Stengel and Forges [26] in their landmark paper on extensive-form correlation. In particular, they show that in two-player perfect-recall games without chance moves, $\Xi$ coincides with a particular polytope $\mathcal{V}$—which we call the *von Stengel-Forges polytope*—whose description only uses a polynomial number of linear constraints, which are "probability-mass-conserving" constraints:

$$
\mathcal{V} := \left\{ \boldsymbol{v} \in \mathbb{R}_{\geq 0}^{|\Sigma_1 \bowtie \Sigma_2|} : \begin{array}{l} \bullet \ v[\varnothing, \varnothing] = 1 \\ \bullet \ \sum_{a \in A_{I_1}} v[(I_1, a), \sigma_2] = v[\sigma(I_1), \sigma_2] \quad \forall I_1 \in \mathcal{I}_1, \sigma_2 \in \Sigma_2 \ \text{s.t.} \ I_1 \bowtie \sigma_2 \\ \bullet \ \sum_{a \in A_{I_2}} v[\sigma_1, (I_2, a)] = v[\sigma_1, \sigma(I_2)] \quad \forall I_2 \in \mathcal{I}_2, \sigma_1 \in \Sigma_1 \ \text{s.t.} \ \sigma_1 \bowtie I_2 \end{array} \right\}.
\tag{1}
$$

The polytope $\mathcal{V}$ is well defined in every game. However, the equality $\Xi = \mathcal{V}$ was known to hold only in two-player games without chance moves. In more general games, it is only known that $\Xi \subseteq \mathcal{V}$. The main contribution of our paper is to show that the equality $\Xi = \mathcal{V}$ holds in significantly more general games than two-player games without chance moves. We will isolate a condition, which we coin *triangle freeness*, that is sufficient for $\Xi = \mathcal{V}$ to hold. We also show that all two-player games where all chance moves are public (including two-player games without chance moves) are triangle free.

## 3 Scaled-Extension-Based Structural Decomposition for $\mathcal{V}$

Farina et al. [14] recently showed that in two-player games without chance moves, a particular structural decomposition theorem holds for the von Stengel-Forges polytope $\mathcal{V}$. At the core of their decomposition is a convexity-preserving operation, *scaled extension*, defined as follows.

**Definition 1** ([14]). *Let $\mathcal{X}$ and $\mathcal{Y}$ be nonempty, compact and convex sets, and let $h : \mathcal{X} \to \mathbb{R}_{\geq 0}$ be a nonnegative affine real function. The* scaled extension *of $\mathcal{X}$ with $\mathcal{Y}$ via $h$ is defined as the set*

$$
\mathcal{X} \overset{h}{\triangleleft} \mathcal{Y} := \{(\boldsymbol{x}, \boldsymbol{y}) : \boldsymbol{x} \in \mathcal{X}, \ \boldsymbol{y} \in h(\boldsymbol{x})\mathcal{Y}\}.
$$

Specifically, they show that in two-player games without chance moves, $\mathcal{V}$ admits a decomposition of the form $\mathcal{V} = \{1\} \triangleleft^{h_1} \mathcal{X}_1 \triangleleft^{h_2} \mathcal{X}_2 \triangleleft^{h_3} \cdots \triangleleft^{h_n} \mathcal{X}_n$, where each of the sets $\mathcal{X}_i$ is either the singleton set $\{1\}$, or a probability simplex $\Delta^{s_i} := \{\boldsymbol{x} \in \mathbb{R}_{\geq 0}^{s_i} : \|\boldsymbol{x}\|_1 = 1\}$ for some appropriate dimension $s_i$.

In this section, we significantly extend their result. As we will show, an analogous scaled-extension-based decomposition of $\mathcal{V}$ exists in far more general games than those without chance moves. In particular, in Section 3.1 we isolate a condition on the information structure of the game—which we coin *triangle freeness*—that guarantees existence of a scaled-extension-based decomposition. Then, we will present an algorithm for computing such a decomposition, that is, finding the $h_i$ functions and sets $\mathcal{X}_i$. Since our full algorithm is rather intricate, we start by giving three examples of increasing complexity that capture the main intuition behind our structural decomposition routine.

**First example**    The first example is shown in the first column of Figure 2. The game starts with a chance node, where two outcomes (say, heads or tails) are possible. After observing the outcome of the chance node, Player 1 chooses between two actions (say, the "left" and the "right" action). The choice as to whether to play the left or the right action can be different based on the observed chance outcome. After Player 1 has played their action, Player 2 has to pick whether to play their left or right action—however, Player 2 does not observe the chance outcome nor Player 1's action. The chance outcome is not observed by Player 2, so, this is not a public-chance game.

The only information set C for Player 2 is connected to both information sets (denoted A and B in Figure 2) of Player 1, so, all sequence pairs $(\sigma_1, \sigma_2) \in \Sigma_1 \times \Sigma_2$ are relevant. Since Player 2 only has one information set, it is easy to incrementally generate the von Stengel-Forges polytope. First, the fixed value 1 is assigned to $v[\varnothing, \varnothing]$ (step ① in the fill-in order). Then, this value is split arbitrarily into the two (non-negative) entries $v[\varnothing, 1], v[\varnothing, 2]$ so that $v[\varnothing, 1] + v[\varnothing, 2] = v[\varnothing, \varnothing]$ in accordance with the von Stengel-Forges constraints. This operation can be expressed using scaled

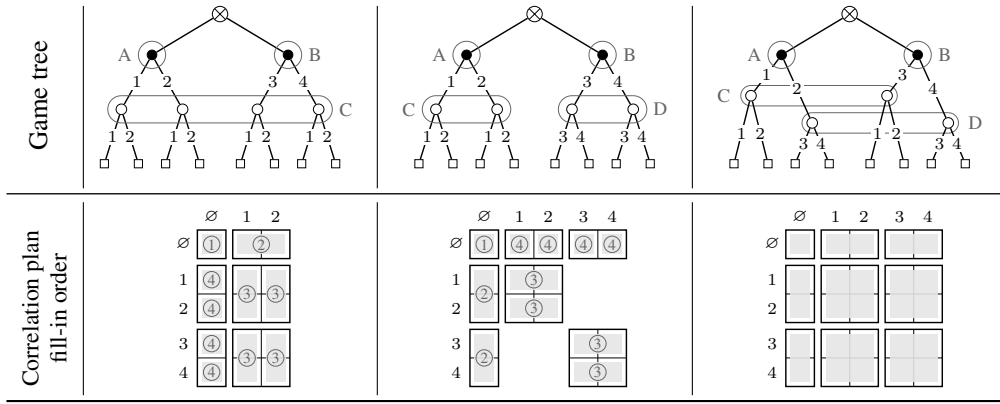

Figure 2: Three examples of extensive-form games with increasingly complex information partitions. Crossed nodes belong the chance player, black round nodes belong to Player 1, white round nodes belong to Player 2, gray round sets define information sets, white squares denote terminal states. The numbers along the edges define concise names for sequences.

extension as $\{(v[\varnothing, \varnothing], v[\varnothing, 1], v[\varnothing, 2])\} = \{1\} \triangleleft^h \Delta^2$, where $h$ is the identity function (step ② in the fill-in order). Then, $v[\varnothing, 1]$ is further split into $v[1,1] + v[2,1] = v[\varnothing, 1]$ and $v[3,1] + v[4,1] = v[\varnothing, 1]$, while $v[\varnothing, 2]$ is split into $v[1,2] + v[2,2] = v[\varnothing, 2]$ and $v[3,2] + v[4,2] = v[\varnothing, 2]$ (step ③ of the fill-in order). These operations can be expressed as scaled extensions with $\Delta^2$. Now that the eight entries $v[\sigma_1, \sigma_2]$ for $\sigma_1 \in \{1,2,3,4\}, \sigma_2 \in \{1,2\}$ have been filled out, we fill in $v[\sigma_1, \varnothing]$ for all $\sigma_1 \in \{1,2,3,4\}$ in accordance with the von Stengel-Forges constraint $v[\sigma_1, \varnothing] = v[\sigma_1, 1] + v[\sigma_1, 2]$ (step ④). In this step, we are not splitting any values, but rather we are summing already-filled-in entries in $v$ to form new entries. Specifically, we can extend the set of partially-filled-in vectors $\boldsymbol{v} = (v[\varnothing, \varnothing], v[\varnothing, 1], v[\varnothing, 2], v[1,1], v[2,1], v[3,1], v[4,1], v[1,2], v[2,2], v[3,2], v[4,2])$ with the new entry $v[1, \varnothing]$ by using the scaled extension operation $\{\boldsymbol{v}\} \triangleleft^h \{1\}$ where $h$ is the (linear) function that extracts the sum $v[\sigma_1, 1] + v[\sigma_1, 2]$ from $\boldsymbol{v}$. By doing so, we have incrementally filled in all entries in $\boldsymbol{v}$. Furthermore, by construction, we have that all von Stengel-Forges constraints $v[\sigma_1, \varnothing] = v[\sigma_1, 1] + v[\sigma_1, 2]$ ($\sigma_1 \in \{\varnothing, 1, 2, 3, 4\}$) and $v[\varnothing, \sigma_2] = v[1, \sigma_2] + v[2, \sigma_2] = v[3, \sigma_2] + v[4, \sigma_2]$ ($\sigma_2 \in \{1, 2\}$) must hold. So, the only two von Stengel-Forges constraints that we have ignored and might potentially be violated are $v[\varnothing, \varnothing] = v[1, \varnothing] + v[2, \varnothing]$ and $v[\varnothing, \varnothing] = v[3, \varnothing] + v[4, \varnothing]$. This concern is quickly resolved by noting that those constraints are implied by the other ones that we satisfy. In particular, by construction we have $v[1, \varnothing] + v[2, \varnothing] = (v[1,1] + v[1,2]) + (v[2,1] + v[2,2]) = (v[1,1] + v[2,1]) + (v[1,2] + v[2,2]) = v[\varnothing, 1] + v[\varnothing, 2] = v[\varnothing, \varnothing]$, and an analogous statement holds for $v[3, \varnothing] + v[4, \varnothing]$. So, all constraints hold and the scaled-extension-based decomposition is finished.

**Remark 1.** *An approach that would start by splitting $v[\varnothing, \varnothing]$ into $v[1, \varnothing] + v[2, \varnothing] = v[\varnothing, \varnothing]$ and $v[3, \varnothing] + v[4, \varnothing] = v[\varnothing, \varnothing]$, thereby inverting the order of fill-in steps ④ and ②, would fail. Indeed, after filling $v[\sigma_1, \sigma_2]$ for all $\sigma_1 \in \{1, 2, 3, 4\}, \sigma_2 \in \{1, 2\}$), there would be no clear way of guaranteeing that $v[1,1] + v[2,1] = v[3,1] + v[4,1] (= v[\varnothing, 1])$.*

**Second example** We now consider a variation of the game from the first example, where Player 2 observes the chance outcome but not the actions selected by Player 1. This game, shown in the middle column of Figure 2, has *public* chance moves, because the chance outcome is observed by all players. In this game, not all pairs of information sets are connected. In fact, only $(A, C)$ and $(B, D)$ are connected information set pairs. Correspondingly, entries such as $v[1,3], v[4,2]$, and $v[2,4]$ are not defined in the correlation plans for the game. This observation is crucial, and will set apart this example from the next one. To fill in any correlation plan, we can start by splitting $v[\varnothing, \varnothing]$ into $v[1, \varnothing] + v[2, \varnothing] = v[\varnothing, \varnothing]$ and $v[3, \varnothing] + v[4, \varnothing] = v[\varnothing, \varnothing]$ (fill-in step ② in the figure). Both operations can be expressed as a scaled extension of partially-filled-in vectors with $\Delta^2$, scaled by the affine function that extracts $v[\varnothing, \varnothing] = 1$ from the partially-filled-in correlation plans. Then, we further split those values into entries $v[\sigma_1, 1] + v[\sigma_1, 2] = v[\sigma_1, \varnothing]$ for $\sigma_1 \in \{1, 2\}$ in accordance with the von Stengel-Forges constraint. Similarly, we will in $v[\sigma_1, 3], v[\sigma_1, 4]$ for $\sigma_1 \in \{3, 4\}$ in accordance with the constraint $v[\sigma_1, 3] + v[\sigma_1, 4] = v[\sigma_1, \varnothing]$ for $\sigma_1 \in \{1, 2\}$ (fill-in step ③). Finally, we recover the values of $v[\varnothing, \sigma_2]$ for $\sigma_2 \in \{1, 2, 3, 4\}$ with a scaled extension with the singleton set $\{1\}$ as discussed in the previous example. Again, it can be checked that despite the fact that we ignored the constraints $v[\varnothing, 1] + v[\varnothing, 2] = v[\varnothing, \varnothing]$ and $v[\varnothing, 3] + v[\varnothing, 4] = v[\varnothing, \varnothing]$, those constraints are automatically satisfied

by constuction. In this case, we were able to sidestep the issue raised in Remark 1 because of the particular connection between the information sets.

**Third example** Finally, we propose a third example in the third column of Figure 2. It is a variation of the first example, where Player 2 now observes Player 1's action but *not* the chance outcome. The most significant difference with the second example is that the information structure of the game is now such that all pairs of information sets of the players are connected. Hence, the problem raised in Remark 1 cannot be avoided. Our decomposition algorithm cannot handle this example.

## 3.1 A Sufficient Condition for the Existence of a Scaled-Extension-Based Decomposition

The third example in the previous section highlights an unfavorable situation in which our decomposition attempt based on incremental generation of the correlation plan. In order to codify all situations in which that issue does not arise, we introduce the concept of rank of an information set.

**Definition 2.** *Let $i \in \{1, 2\}$ be one player, and let $-i$ denote the other player. Furthermore, let $I \in \mathcal{I}_i$ and $\sigma \in \Sigma_{-i}$. The $\sigma$-rank of $I$ is the cardinality of the set $\{J \in \mathcal{I}_{-i} : J \rightleftharpoons I, \sigma(J) = \sigma\}$.*

The issue in Remark 1 can be stated in terms of the ranks. Consider a relevant sequence pair $(\sigma_1, \sigma_2) \in \Sigma_1 \bowtie \Sigma_2$ and two connected information sets $I_1 \rightleftharpoons I_2$ such that $\sigma(I_1) = \sigma_1, \sigma(I_2) = \sigma_2$. If the $\sigma_1$-rank of $I_2$ and the $\sigma_2$-rank of $I_1$ are both greater than 1, the issue cannot be avoided and the decomposition will fail. For example, in the third example, where our decomposition fails, all information sets have $\varnothing$-rank 2. We prove that such situations cannot occur, provided the game satisfies the following condition, which can be verified in polynomial time in the size of the EFG.

**Definition 3** (Triangle-freeness)**.** *A two-player extensive-form game is* triangle-free *if, for any choice of two distinct information sets $I_1, I_2 \in \mathcal{I}_1$ such that $\sigma(I_1) = \sigma(I_2) = \sigma_1$ and two distinct information sets $J_1, J_2 \in \mathcal{I}_2$ such that $\sigma(J_1) = \sigma(J_2) = \sigma_2$, it is never the case that $I_1 \rightleftharpoons J_1 \wedge I_2 \rightleftharpoons J_2 \wedge I_1 \rightleftharpoons J_2$.*

In Theorem 1 we show that games with public chance (which includes games with no chance moves at all) always satisfy the triangle-freeness condition of Definition 3.

**Theorem 1.** *A two-player extensive-form game with public chance moves is triangle-free.*

However, not all triangle-free games must have public chance nodes. For example, the leftmost game in Figure 2 is triangle-free, but in that game the chance outcome is not public to Player 2. So, our results apply more broadly than games with public chance moves.

## 3.2 Computation of the Decomposition

We present our algorithm following the same structure as [14]. Like theirs, our algorithm consists of a recursive function, DECOMPOSE. It takes three arguments: (i) a sequence pair $(\sigma_1, \sigma_2) \in \Sigma_1 \bowtie \Sigma_2$, (ii) a subset $\mathcal{S}$ of the set of all relevant sequence pairs, and (iii) a set $\mathcal{D}$ where only the entries indexed by the elements in $\mathcal{S}$ have been filled in. The decomposition for the whole von Stengel-Forges polytope $\mathcal{V}$ is computed by calling DECOMPOSE$((\varnothing, \varnothing), \{(\varnothing, \varnothing)\}, \{(1)\})$—this corresponds to the starting situation in which only the entry $v[\varnothing, \varnothing]$ has been filled in (denoted as fill-in step ① in Figure 2). Each call to DECOMPOSE returns a pair $(\mathcal{S}', \mathcal{D}')$ of updated indices and partial vectors, to reflect the new entries that were filled in during the call.

DECOMPOSE$((\sigma_1, \sigma_2), \mathcal{S}, \mathcal{D})$ operates as follows (we denote with $-i$ the opponent for Player $i$):

1. Let $\mathcal{J}_i := \{I \in \mathcal{I}_i : I \bowtie \sigma_{-i}, \sigma(I) = \sigma_i\}$ for all $i \in \{1, 2\}$, and $\mathcal{J}^* \leftarrow \emptyset$.
2. For each $(I_1, I_2) \in \mathcal{J}_1 \times \mathcal{J}_2$ such that $I_1 \rightleftharpoons I_2$, if the $\sigma_2$-rank of $I_1$ is greater than or equal to the $\sigma_1$-rank of $I_2$, we update $\mathcal{J}^* \leftarrow \mathcal{J}^* \cup \{I_1\}$. Else, we update $\mathcal{J}^* \leftarrow \mathcal{J}^* \cup \{I_2\}$.
3. For each $i \in \{1, 2\}$ and $I \in \mathcal{J}_i$ such that the $\sigma_{-i}$-rank of $I$ is 0, do $\mathcal{J}^* \leftarrow \mathcal{J}^* \cup \{I\}$.
4. For each $I \in \mathcal{J}^*$: (Below we assume that $I \in \mathcal{I}_1$, the other case is symmetrical)
   (a) Fill in all entries $\{v[(I, a), \sigma_2] : a \in A_I\}$ by splitting $v[\sigma_1, \sigma_2]$. This can be expressed using a scaled extension operation as $\mathcal{D} \leftarrow \mathcal{D} \triangleleft^h \Delta^{|A_I|}$ where $h$ extracts $v[\sigma_1, \sigma_2]$ from any partially-filled-in vector.
   (b) Update $\mathcal{S} \leftarrow \mathcal{S} \cup \{((I, a), \sigma_2)\}$ to reflect that the entries corresponding to $(I, a) \bowtie \sigma_2$ have been filled in.
   (c) For each $a \in A_I$ we assign $(\mathcal{S}, \mathcal{D}) \leftarrow$ DECOMPOSE$(((I, a), \sigma_2), \mathcal{S}, \mathcal{D})$. End for.
   (d) Let $\mathcal{K} := \{J \in \mathcal{I}_2 : I \rightleftharpoons J\}$. For all $J \in \mathcal{I}_2$ such that $\sigma(J) \succeq (J', a')$ for some $J' \in \mathcal{K}, a' \in A_{J'}$:

- If $I \rightleftharpoons J$, then for all $a \in A_J$ we fill in the sequence pair $v[\sigma_1, (J, a)]$ by assigning its value in accordance with the von Stengel-Forges constraint $v[\sigma_1, (J, a)] = \sum_{a^* \in A_{I^*}} v[(I^*, a^*), (J, a)]$ via the scaled extension $\mathcal{D} \leftarrow \mathcal{D} \triangleleft^h \{1\}$ where the linear function $h$ maps a partially-filled-in vector to the value of $\sum_{a^* \in A_{I^*}} v[(I^*, a^*), (J, a)]$. Since this is done for all $a \in A_J$, automatically $\sum_{a \in A_J} v[\sigma_1, (J, a)] = v[\sigma_1, \sigma_2]$, and we can safely ignore the latter constraint.
  - Otherwise, we fill in the entries $\{v[\sigma_1, (J, a)] : a \in A_J\}$, by splitting the value $v[\sigma_1, \sigma(J)]$. In this case, we let $\mathcal{D} \leftarrow \mathcal{D} \triangleleft^h \Delta^{|A_J|}$ where $h$ extracts the entry $v[\sigma_1, \sigma(J)]$ from a partially-filled-in vector in $\mathcal{D}$.
5. At this point, all the entries corresponding to indices $\tilde{S} = \{(\sigma'_1, \sigma'_2) : \sigma'_1 \succeq \sigma_1, \sigma'_2 \succeq \sigma_2\}$ have been filled in, and we return $(S \cup \tilde{S}, \mathcal{D})$.

Pseudocode is available in Appendix A. The above algorithm formalizes and generalizes the first two examples of Figure 2. For example, step ② of the fill-in order in either example is captured in Step 4(a), while fill-in step ③ corresponds to Step 4(c). Finally, fill-in step ④ corresponds to Step 4(d).

Compared to the decomposition algorithm by Farina et al. [14], our branching steps (Step 4) are significantly more intricate. This is because, compared to their setting (that is, two-player games without chance moves) where at least one player has at most one information set with rank strictly greater than one, we have to account for multiple information sets with rank greater than one. Since two-player games without chance moves are a special case of two-player games with public chance moves, our algorithm completely subsumes that of Farina et al. [14].

A proof of correctness for the algorithm is in Appendix A. In particular, the following holds.

**Theorem 2.** *The von Stengel-Forges polytope $\mathcal{V}$ of a two-player perfect-recall triangle-free EFG can be expressed via a sequence of scaled extensions with simplexes and singleton sets:*

$$\mathcal{V} = \{1\} \overset{h_1}{\triangleleft} \mathcal{X}_1 \overset{h_2}{\triangleleft} \mathcal{X}_2 \overset{h_3}{\triangleleft} \cdots \overset{h_n}{\triangleleft} \mathcal{X}_n, \tag{2}$$

*where, for $i = 1, \dots, n$, either $\mathcal{X}_i = \Delta^{s_i}$ for some simplex dimension $s_i \in \mathbb{N}$, or $\mathcal{X}_i = \{1\}$, and $h_i$ is a linear function. Furthermore, an exact algorithm exists to compute such expression in linear time in the dimensionality of $\mathcal{V}$, and so, in time at most quadratic in the size of the game.*

## 4 Bridging the Gap Between $\mathcal{V}$ and $\Xi$

As noted by von Stengel and Forges [26], the inclusion $\Xi \subseteq \mathcal{V}$ holds trivially in any game. The reverse inclusion, $\Xi \supseteq \mathcal{V}$, was shown for two-player games without chance moves, but no complete characterization as to when that reverse inclusion holds was known before our paper. In Theorem 3, we contribute a new connection between the reverse inclusion $\Xi \supseteq \mathcal{V}$ and the integrality of the vertices of the von Stengel-Forges polytope (all proofs are in Appendix B).

**Theorem 3.** *Let $\Gamma$ be a two-player perfect-recall extensive-form game, let $\mathcal{V}$ be its von Stengel-Forges polytope, and let $\Xi$ be its polytope of correlation plans. Then, $\Xi = \mathcal{V}$ if and only if all vertices of $\mathcal{V}$ have integer $\{0, 1\}$ coordinates.*

As it turns out, the scaled-based decomposition of $\mathcal{V}$ can be used to conclude the integrality of the vertices of $\mathcal{V}$, by leveraging the following analytical result about the scaled extension operation.

**Lemma 1.** *Let $\mathcal{X}, \mathcal{Y}$, and $h$ be as in Definition 1. If $\mathcal{X}$ is a convex polytope with vertices $\{\boldsymbol{x}_1, \dots, \boldsymbol{x}_n\}$, and $\mathcal{Y}$ is a convex polytope with vertices $\{\boldsymbol{y}_1, \dots, \boldsymbol{y}_m\}$, then $\mathcal{X} \triangleleft^h \mathcal{Y}$ is a convex polytope whose vertices are a nonempty subset of $\{(\boldsymbol{x}_i, h(\boldsymbol{x}_i)\boldsymbol{y}_j) : i \in \{1, \dots, n\}, j \in \{1, \dots, m\}\}$.*

In particular, by applying Lemma 1 inductively on the structure of the scaled-extension-based structural decomposition of $\mathcal{V}$, we obtain the following theorem.

**Theorem 4.** *Let $\mathcal{V}$ be the von Stengel-Forges polytope of a two-player triangle-free game (Definition 3). All vertices of $\mathcal{V}$ have integer $\{0, 1\}$ coordinates.*

Finally, combining Theorem 4 and Theorem 3, we obtain the central theorem of this paper.

**Theorem 5.** *In a two-player perfect-recall extensive-form game that satisfies the triangle-freeness condition (Definition 3), the polytope of correlation plans coincides with the von Stengel-Forges polytope. Consequently, an optimal EFCE, EFCCE, or NFCCE can be computed in polynomial time (in the size of the input extensive-form game) in two-player triangle-free games.*

A consequence of $\mathcal{V} = \Xi$ is that the linear programs for EFCE [26], EFCCE [15] and NFCCE [15]—originally formulated for two-player games without chance moves only—hold verbatim for any triangle-free game. So, an optimal EFCE, EFCCE, and NFCCE can be computed in polynomial time as the solution of those linear programs. Furthermore, the scaled-extension-based decomposition for triangle-free games (Section 3) can be combined with the *scaled extension regret circuit* introduced by Farina et al. [14, 12] to construct a scalable regret minimization algorithm for $\mathcal{V} = \Xi$. That, in turn, can be used to compute an EFCE, EFCCE, and NFCCE in large triangle-free games that are too large for traditional linear programming methods.

## 5 Experimental Evaluation

We implemented the scaled-extension-based decomposition routine of Section 3. We test our decomposition algorithm for triangle-free games on Goofspiel [25], a popular benchmark game in computational game theory. In Goofspiel, each player has a personal deck of cards made of $k$ different ranks (from $1$ to $k$). A third deck (the "prize" deck) is shuffled and put face down on the board at the beginning of the game. In each turn, the topmost card from the prize deck is *publicly* revealed. Then, each player privately picks a card from their hand—this card acts as a bid to win the card that was just revealed from the prize deck. The player that bids the highest wins the prize card. We use an established tie-breaking rule: the prize card is discarded if the players' bids are equal. Furthermore, we adopt the convention that only the winner is revealed, but not the bids, in accordance with prior computational game theory literature [21, 20]. The players' scores are computed as the sum of the values of the prize cards they have won. Because of the tie-breaking rule, Goofspiel is a general-sum game. Furthermore, since all chance outcomes are public, it is a triangle-free game.

In Figure 3(left) we report the performance of our decomposition routine for $k = 3, 4, 5$, both in terms of number of scaled extension operations required in the decomposition (Theorem 2) and of runtime of our single-threaded implementation, as well as the dimensions of the games. The runtime was averaged over 100 independent runs. Our decomposition algorithm performs well, and is able to scale to the largest game (Goofspiel with $k = 5$ ranks, which has $3.6 \times 10^7$ relevant sequence pairs).

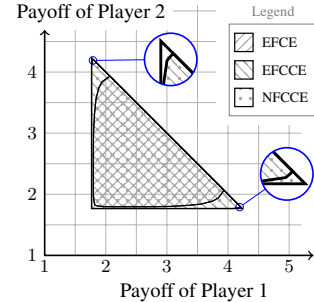

| Deck size | Information sets | | Sequences | | Decomposition | |
|---|---|---|---|---|---|---|
| | $|\mathcal{I}_1 \cup \mathcal{I}_2|$ | $|\mathcal{I}_1 \rightleftharpoons \mathcal{I}_2|$ | $|\Sigma_1 \cup \Sigma_2|$ | $|\Sigma_1 \bowtie \Sigma_2|$ | Num $\lhd^h$ | Runtime |
| 3 ranks | $4.3 \times 10^2$ | $1.1 \times 10^3$ | $5.2 \times 10^2$ | $3.3 \times 10^3$ | $2.9 \times 10^3$ | 2ms |
| 4 ranks | $1.7 \times 10^4$ | $8.1 \times 10^4$ | $2.1 \times 10^4$ | $2.7 \times 10^5$ | $2.4 \times 10^5$ | 1.1s |
| 5 ranks | $1.2 \times 10^6$ | $1.1 \times 10^7$ | $1.4 \times 10^6$ | $3.6 \times 10^7$ | $3.2 \times 10^7$ | 43.8s |

Figure 3: (Left) Dimensions of games and runtime of decomposition algorithm (Theorem 2). (Right) Payoffs that can be reached using an EFCE, EFCCE, or NFCCE in $3$-rank Goofspiel.

We also implemented the linear programming formulation of EFCE described by von Stengel and Forges [26] together with our characterization of $\Xi$ (Theorem 5), and the scalable regret minimization algorithm of Farina et al. [14]. We use the Gurobi commercial linear programming solver to solve the linear program formulation.[1] As the game size increases, the barrier algorithm is the only algorithm that can solve the linear program. However, even that quickly becomes impractical. In the largest game, Gurobi uses roughly 200GB of memory, spends approximately 90 minutes to precondition the linear program, and requires slightly more than 20 minutes to perform each iteration of the barrier method using 30 threads. The regret minimization scales significantly more favorable in the large game. It requires roughly 6 seconds per iteration, and reaches $10^{-2}$ infeasibility in 4 minutes, $10^{-3}$ infeasibility (defined as how incentive-incompatible the computed correlation plan is, measured as the difference in value that each player would gain by optimally deviating from any recommendation at any information set in the game) in 12 minutes, and $10^{-4}$ infeasibility in 110 minutes. Additional

data about the experiment is available in Appendix C. This extends prior findings that the regret minimization algorithm is more scalable—both in terms of run time and memory—than the linear programming approach in large games to triangle-free games as well [14].

In Figure 3(right) we used the characterization $\Xi = \mathcal{V}$ to compute the set of all payoffs that can be reached by an EFCE, EFCCE, or NFCCE in 3-rank Goofspiel. The sets are highly non-trivial, and reinforce the observation that the behaviors and incentives that can be induced through extensive-form correlation are subtle and complex [13]. The sets of reachable payoff vectors were computed as follows. We considered 1000 unit vectors $(\alpha, \beta)$ equally spaced on the unit two-dimensional ball. For each choice $(\alpha, \beta)$, we computed the values $v_{\alpha,\beta}^{\text{efce}}, v_{\alpha,\beta}^{\text{efcce}}, v_{\alpha,\beta}^{\text{nfcce}}$ of an EFCE, EFCCE, and NFCCE of the game, respectively, that maximizes the objective $\alpha \times$ payoff of Player $1 + \beta \times$ payoff of Player 2. using the linear programming formulation of EFCE made possible by Theorem 5 solved by Gurobi. Each of these values shows that the set of all payoffs that can be induced by EFCE (resp., EFCCE, NFCCE) must satisfy the inequality $\alpha \times$ payoff of Player $1 + \beta \times$ payoff of Player $2 \le v_{\alpha,\beta}^{\text{efce}}$ (resp., $v_{\alpha,\beta}^{\text{efcce}}, v_{\alpha,\beta}^{\text{nfcce}}$). This defines an outer description of the polytope of reachable payoffs for each solution concept. Taking the intersection of all those inequalities yields the polytopes shown in Figure 3(right). We used *Polymake*, a library for computational polyhedral geometry [16, 2], to compute that intersection. To our knowledge, we are the first to fully characterize the set of correlated solution concepts in Goofspiel.

# 6  Conclusions and Future Directions

We showed that an optimal extensive-form correlated equilibrium, extensive-form coarse correlated equilibrium, and normal-form coarse correlated equilibrium can be computed in polynomial time in two-player perfect-recall games that satisfy a certain *triangle-freeness* condition that we introduced and that can be checked in polynomial time. To show that such equilibria can be found in polynomial time, we gave and combined several results that may be of independent interest: (1) the existence of a scaled-extension-based structural decomposition for the von Stengel-Forges polytope of the game, (2) a characterization of when the von Stengel-Forges polytope coincides with the polytope of correlation plans, and (3) a result about the integrality of the vertices of the von Stengel-Forges polytope in triangle-free games.

Several questions remain open about correlation in extensive-form games, both from a complexity point of view, and from a practical point of view. In theory, the regret minimization algorithm of Farina et al. [14] can be used to compute EFCE, EFCCE, and NFCCE with a given lower bound on a given linear objective. Hence, to optimize a given objective one could perform a binary search on the optimal objective value by running the regret minimization method several times with different lower bounds. No experimental evaluation of that is currently available in the literature. It is also unknown how the regret minimization algorithm and the linear-programming-based algorithm used in Section 5 (which can compute optimal correlated solution concepts) compare in practice to other methods to compute *one* correlated solution concept, such as the algorithms by Dudík and Gordon [11], Huang and von Stengel [18], and more recently Celli et al. [10].

# Acknowledgments

This material is based on work supported by the National Science Foundation under grants IIS-1718457, IIS-1617590, IIS-1901403, and CCF-1733556, and the ARO under awards W911NF-17-1-0082 and W911NF2010081. Gabriele Farina is supported by a Facebook fellowship.

# Broader Impact

Correlated solution concepts have many advantages. First, they enable incentive-compatible coordination of agents. Such coordination is achieved via incentives, rather than forcing: mediators in correlated solution concepts are only able to recommend behavior, but not force it. So, it is up to the mediator to come up with a correlated distribution of recommendations such that no agent has incentive to deviate from the recommendations. Second, in some general-sum interactions these solution concepts are known to enable significantly higher social welfare than Nash equilibrium, while at the same time sidestepping some of the other shortcomings of Nash equilibrium (for example,

some equilibrium selection issues that make Nash equilibrium less appealing as a prescriptive tool for rational behavior).

In this paper, we are particularly interested in *optimal* correlated equilibria. In other words, our technology can empower the system designer (mediator) to select, among the infinite number of correlated equilibria of the game, one that maximizes a given objective. For example, this technology could be used to find correlated equilibria that maximize the sum of utilities of the players, potentially leading to higher societal good. However, like most technology, our technology has potential for abuse. If used maliciously, the ability to select particular correlated equilibria could be used to *minimize* social welfare, maximize only one of the agent's utility, or minimize all others' utilities—thereby furthering existing inequality or creating new inequality.

## Footnotes

[1]At the time of writing, Gurobi is freely available for academic use. Free and open-source linear programming solvers such as GLPK could be used instead, though they tend to be slower and less numerically stable than commercial solutions.

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
