[Supplementary Material]

# A  Scaled-Extension-Based Structural Decomposition for $\mathcal{V}$

## A.1  Triangle-Freeness

**Lemma 2.** *Consider a triangle-free game, let $(\sigma_1, \sigma_2) \in \Sigma_1 \bowtie \Sigma_2$, and let $I_1 \rightleftharpoons I_2$ be such that $\sigma(I_1) = \sigma_1, \sigma(I_2) = \sigma_2$. Then, at most one between the $\sigma_1$-rank of $I_2$ and the $\sigma_2$-rank of $I_1$ is strictly larger than $1$.*

*Proof.* The results follows almost immediately from the definition of triangle-freeness. We prove the statement by contradiction. Let $(\sigma_1, \sigma_2) \in \Sigma_1 \bowtie \Sigma_2$ be a relevant sequence pair, and let information sets $I_1 \in \mathcal{I}_1, I_2 \in \mathcal{I}_2$ be such that $\sigma(I_1) = \sigma_1, \sigma(I_2) = \sigma_2$. Furthermore, assume that the $\sigma_1$-rank of $I_2$ is greater than $1$, and at the same time the $\sigma_2$-rank of $I_1$ is greater than $1$. Since the $\sigma_2$-rank of $I_1$ is greater than $1$, there exists an information set $I_2' \in \mathcal{I}_2, \sigma(I_2') = \sigma_2$, distinct from $I_2$, such that $I_1 \rightleftharpoons I_2'$. Similarly, because the $\sigma_1$-rank of $I_2$ is greater than $1$, there exists an information set $I_1' \in \mathcal{I}_1, \sigma(I_1') = \sigma_1$, distinct from $I_1$, such that $I_1' \rightleftharpoons I_2$. But then, we have found $I_1, I_1' \in \mathcal{I}_1$ and $I_2', I_2 \in \mathcal{I}_2$ such that $\sigma(I_1) = \sigma(I_2) = \sigma_1, \sigma(I_2') = \sigma(I_2) = \sigma_2$ such that $I_1 \rightleftharpoons I_2', I_1' \rightleftharpoons I_2$, and $I_1 \rightleftharpoons I_2$. So, the game is *not* triangle-free, contradiction. $\square$

**Theorem 1.** *A two-player extensive-form game with public chance moves is triangle-free.*

*Proof.* For contradiction, let $I_1, I_2$ be two distinct information sets for Player 1 such that $\sigma(I_1) = \sigma(I_2)$, let $J_1, J_2$ be two distinct information sets for Player 2 such that $\sigma(J_1) = \sigma(J_2)$, and assume that $I_1 \rightleftharpoons J_1, I_2 \rightleftharpoons J_2, I_1 \rightleftharpoons J_2$. By definition of connectedness, there exist nodes $u \in I_1, v \in J_1$ such that $v$ is on the path from the root to $u$, or *vice versa*. Similarly, there exist nodes $u' \in I_2, v' \in J_2$ such that $u'$ is on the path from the root to $v'$, or *vice versa*. Let $w$ be the lowest common ancestor of $u$ and $u'$. It is not possible that $w = u$ or $w = u'$, because otherwise the parent sequences of $I_1$ and $I_2$ would be different. So $w$ must be a strict ancestor of both $u$ and $u'$, and $u$ and $u'$ must be reached using *different* edges at $w$. Therefore, node $w$ cannot belongs to Player 1, or otherwise it again would not be true that $\sigma(I_1) = \sigma(I_2)$. So, there are only two possible cases: either $w$ belongs to Player 2, or it belongs to the chance player. We break the analysis accordingly.

- **First case:** $w$ belongs to Player 2. From above, we know that $u$ and $u'$ are reached by following different branches at $w$. So, if both $v$ and $v'$ were strict descendants of $w$, they would need to be on two different branches of $w$ (because they are connected to $u$ and $u'$ respectively), violating the condition $\sigma(J_1) = \sigma(J_2)$. So, at least one between $v$ and $v'$ is on the path from the root to $w$ (inclusive). But then either $v$ is an ancestor of $v'$, or *vice versa*. Either case violates the hypothesis that $\sigma(J_1) = \sigma(J_2)$.

- **Second case:** $w$ belongs to the chance player. If any between $v$ and $v'$ is an ancestor of $w$, then necessarily either $v$ is an ancestor of $v'$, or $v'$ is an ancestor of $v$. Either case violates the condition $\sigma(J_1) = \sigma(J_2)$. So, both $v$ and $v'$ must be descendants of $w$. Because $v$ is on the path from the root to $u$ (or *vice versa*), and $v'$ is on the path from the root to $u'$ (or *vice versa*), then necessarily $u, v$ and $u', v'$ are on two different branches of the chance node $w$. To fix names, call $a$ the action at $w$ that must be taken to (eventually) reach $u$ and $v$, and let $b$ be the action that must be taken to (eventually) reach $u'$ and $v'$. Now, we use the hypothesis that $I_1 \rightleftharpoons J_2$, that is, there exists $u'' \in I_1, v'' \in J_2$ such that $u''$ is on the path from the root to $v''$ or *vice versa*. Assume that $u''$ is on the path from the root to $v''$. Since $u''$ belongs to the same information set as $u$ (that is, $I_1$), and since chance is public by hypothesis, then Player 1, when acting at $u$ and $u''$, must have observed action $a$ at $w$. In other words, the path from the root to $u''$ must pass through action $a$ at $w$. But then, using the fact that $u''$ is on the path from the root to $v''$, this means that the path from the root to $v''$ passes through action $a$. However, the path from the root to $v'$ passes through action $b$. Since chance is public, nodes $v'$ and $v''$ cannot be in the same information set, because Player 2 is able to distinguish them by means of the observed chance outcome. We reached a contradiction. The symmetric case where $v''$ is on the path from the root to $u''$ is analogous. $\square$

## A.2  Decomposition Algorithm

In this section, we provide pseudocode for the algorithm presented in Section 3.2. We will use the following conventions:

- Given a player $i \in \{1, 2\}$, we let $-i$ denote the opponent.

- We use the symbol $\sqcup$ to denote disjoint union.
- Given two infosets $I, I' \in \mathcal{I}_i$, we write $I \preceq I'$ if $\sigma(I') \succeq \sigma(I)$. We say that we iterate over a set $\mathcal{I} \subseteq \mathcal{I}_i$ *in top-down order* if, given any two $I, I' \in \mathcal{I}$ such that $I \preceq I'$, $I$ appears before $I'$ in the iteration.
- We use the observation that for all $I \in \mathcal{I}_1$ and $\sigma_2 \in \Sigma_2$, $I \bowtie \sigma_2$ if and only if $(I,a) \bowtie \sigma_2 \; \forall a \in A_I$. (A symmetric statement holds for $I \in \mathcal{I}_2$ and $\sigma_1 \in \Sigma_1$.)

### A.2.1 Two Useful Subroutines

We start by presenting two simple subroutines that capture fill-in step ④ of Figure 2 or equivalently Step 4(d) of Section 3.2. The two subroutines are symmetric and have the role of filling rows and columns of the correlation plans.

---

**Algorithm 1:** FILLOUTROW$((\sigma_1, \sigma_2), I_1, \mathcal{S}, \mathcal{D})$

---

**Preconditions:** $(\sigma_1, \sigma_2) \in \Sigma_1 \bowtie \Sigma_2, I_1 \in \mathcal{I}_1, \sigma(I_1) = \sigma_1, (\sigma_1, \sigma_2) \in \mathcal{S}$

1  **for** $I_2$ *such that* $\sigma(I_2) = \sigma_2$ *and* $\sigma_1 \bowtie I_2$ **do**
2  $\quad$ **if** $I_1 \rightleftharpoons I_2$ **then**
3  $\quad\quad$ **for** $\sigma_2' \in \{(I_2, a) : a \in A_{I_2}\}$ **do**
   $\quad\quad\quad$ ▷ Fill $(\sigma_1, \sigma_2')$ by summing up all entries $\{v[(I_1, a'), \sigma_2'] : a' \in A_{I_1}\}$ in accordance with the von Stengel-Forges constraints
4  $\quad\quad\quad$ $\mathcal{S} \leftarrow \mathcal{S} \sqcup \{(\sigma_1, \sigma_2')\}$
5  $\quad\quad\quad$ $\mathcal{D} \leftarrow \mathcal{D} \triangleleft^h \{1\}$ where $h : \boldsymbol{v} \mapsto \sum_{a' \in A_{I_1}} v[(I_1, a'), \sigma_2']$
6  $\quad$ **else**
   $\quad\quad$ ▷ Fill all $\{v[\sigma_1, (I_2, a)] : a \in A_{I_2}\}$ by splitting $v[\sigma_1, \sigma_2]$ accordance with the von Stengel-Forges constraints
7  $\quad\quad$ $\mathcal{S} \leftarrow \mathcal{S} \sqcup \{(\sigma_1, (I_2, a)) : a \in A_{I_2}\}$
8  $\quad\quad$ $\mathcal{D} \leftarrow \mathcal{D} \triangleleft^h \Delta^{|A_{I_2}|}$ where $h : \boldsymbol{v} \mapsto v[\sigma_1, \sigma_2]$
9  $\quad$ **for** $\sigma_2' \in \{(I_2, a) : a \in A_{I_2}\}$ **do**
10 $\quad\quad$ FILLOUTROW$((\sigma_1, \sigma_2'), I_1)$
11 **return** $(\mathcal{S}, \mathcal{D})$

---

**Algorithm 2:** FILLOUTCOLUMN$((\sigma_1, \sigma_2), I_2, \mathcal{S}, \mathcal{D})$

---

**Preconditions:** $(\sigma_1, \sigma_2) \in \Sigma_1 \bowtie \Sigma_2, I_2 \in \mathcal{I}_2, \sigma(I_2) = \sigma_2, (\sigma_1, \sigma_2) \in \mathcal{S}$

1  **for** $I_1$ *such that* $\sigma(I_1) = \sigma_1$ *and* $\sigma_2 \bowtie I_1$ **do**
2  $\quad$ **if** $I_1 \rightleftharpoons I_2$ **then**
3  $\quad\quad$ **for** $\sigma' \in \{(I_1, a) : a \in A_{I_1}\}$ **do**
   $\quad\quad\quad$ ▷ Fill $(\sigma_1', \sigma_2)$ by summing up all entries $\{v[\sigma_1', (I_2, a')] : a' \in A_{I_2}\}$ in accordance with the von Stengel-Forges constraints
4  $\quad\quad\quad$ $\mathcal{S} \leftarrow \mathcal{S} \sqcup \{(\sigma_1', \sigma_2)\}$
5  $\quad\quad\quad$ $\mathcal{D} \leftarrow \mathcal{D} \triangleleft^h \{1\}$ where $h : \boldsymbol{v} \mapsto \sum_{a' \in A_{I_2}} v[\sigma_1', (I_2, a')]$
6  $\quad$ **else**
   $\quad\quad$ ▷ Fill all $\{v[(I_1, a), \sigma_2] : a \in A_{I_1}\}$ by splitting $v[\sigma_1, \sigma_2]$ accordance with the von Stengel-Forges constraints
7  $\quad\quad$ $\mathcal{S} \leftarrow \mathcal{S} \sqcup \{((I_1, a), \sigma_2) : a \in A_{I_1}\}$
8  $\quad\quad$ $\mathcal{D} \leftarrow \mathcal{D} \triangleleft^h \Delta^{|A_{I_1}|}$ where $h : \boldsymbol{v} \mapsto v[\sigma_1, \sigma_2]$
9  $\quad$ **for** $\sigma' \in \{(I_1, a) : a \in A_{I_1}\}$ **do**
10 $\quad\quad$ FILLOUTCOLUMN$((\sigma_1', \sigma_2), I_2)$
11 **return** $(\mathcal{S}, \mathcal{D})$

---

The following inductive contract will be important for the full algortihm.

**Lemma 3** (Inductive contract for FILLOUTROW). *Suppose that the following preconditions hold when* FILLOUTROW$((\sigma_1, \sigma_2), I_1, \mathcal{S}, \mathcal{D}))$ *is called:*

*(Pre1)* $(\sigma_1, \sigma_2) \in \Sigma_1 \bowtie \Sigma_2$*;*

*(Pre2)* $I_1 \in \mathcal{I}_1$ *is such that* $\sigma(I_1) = \sigma$*;*

*(Pre3)* $\mathcal{S}$ *contains only relevant sequence pairs and* $\mathcal{D}$ *consists of vectors indexed by exactly the indices in* $\mathcal{S}$*;*

*(Pre4)* $(\sigma_1, \sigma_2) \in \mathcal{S}$*, but* $(\sigma_1, \sigma_2') \notin \mathcal{S}$ *for all* $\sigma_2' \succ \sigma_2$*;*

*(Pre5)* For all $a \in I_1$ and $\sigma_2' \succeq \sigma_2$ such that $I_1 \bowtie \sigma_2'$, $((I_1, a), \sigma_2') \in \mathcal{S}$;

*(Pre6)* If $I_1 \bowtie \sigma_2$, all $\boldsymbol{v} \in \mathcal{D}$ satisfy the von Stengel-Forges constraint $v[\sigma_1, \sigma_2] = \sum_{a \in I_1} v[(I_1, a), \sigma_2]$;

*(Pre7)* All $\boldsymbol{v} \in \mathcal{D}$ satisfy the von Stengel-Forges constraints
$$v[(I, a), \sigma(I_2)] = \sum_{a' \in A_{I_2}} v[(I, a), (I_2, a')] \quad \forall a \in I_1, \text{ and } I_2 \in \mathcal{I}_2 : I_1 \bowtie I_2, \sigma(I_2) \succeq \sigma_2.$$

*Then, the sets $(\mathcal{S}', \mathcal{D}')$ returned by the call are such that*

*(Post1)* $\mathcal{S}'$ contains only relevant sequence pairs and $\mathcal{D}'$ consists of vectors indexed by exactly the indices in $\mathcal{S}'$;

*(Post2)* $\mathcal{S}' = \mathcal{S} \sqcup \{(\sigma_1, \sigma_2') : \sigma_2' \succ \sigma_2, \sigma \bowtie \sigma_2'\}$;

*(Post3)* All $\boldsymbol{v} \in \mathcal{D}'$ satisfy the von Stengel-Forges constraints
$$v[\sigma_1, \sigma(I_2)] = \sum_{a' \in A_{I_2}} v[\sigma_1, (I_2, a')] \quad \forall I_2 \in \mathcal{I}_2 : \sigma \bowtie I_2, \sigma(I_2) \succeq \sigma_2$$

and all von Stengel-Forges constraints
$$v[\sigma_1, \sigma_2'] = \sum_{a \in A_{I_1}} v[(I, a), \sigma_2'] \quad \forall \sigma_2' \in \Sigma_2 : \sigma_2' \bowtie I_1, \sigma_2' \succeq \sigma_2.$$

*Proof.* By induction.

- **Base case.** The base case corresponds to $\sigma_2 \in \Sigma_2$ such that no information set $I_2 \in \mathcal{I}_2 : \sigma(I_2) = \sigma_2 \wedge \sigma_1 \bowtie I_2$ exists. In that case, Algorithm 1 returns immediately, so (Post1) holds trivially from (Pre3). Since no $I_2$ such that $\sigma(I_2) = \sigma_2 \wedge \sigma_1 \bowtie I_2$ exists, no $\sigma_2' \succ \sigma_2$ such that $\sigma_1 \bowtie \sigma_2'$ exists, so (Post2) holds as well. The first set of constraints of (Post3) is empty, and the second set reduces to (Pre6).

- **Inductive step.** Suppose that the inductive hypothesis holds when $\sigma_2' \succ \sigma_2$. We will show that it holds when $\sigma_2' = \sigma_2$ as well. In order to use the inductive hypothesis, we first need to check that the preconditions are preserved at the time of the recursive call on Line 10. (Pre1) holds since $\sigma_1 \bowtie I_2$. (Pre2) holds trivially since $\sigma$ does not chance. (Pre3) holds since we are updating $\mathcal{S}$ and $\mathcal{D}$ in tandem on lines 4, 5 and 7, 8. (Pre4) holds since by the time of the recursive call we have only filled in entries $(\sigma_1, \sigma_2')$ where $\sigma_2'$ is an immediate successor of $\sigma_2$. (Pre5) at Line 10 holds trivially, since it refers to a subset of the entries for which the condition held at the beginning of the call. (Pre6) holds because $I_1 \bowtie \sigma_2' \iff I_1 \rightleftharpoons I_2$. Hence, if $I_1 \bowtie \sigma_2'$ then Lines 4 and 5 must have run. (Pre7) at Line 10 holds trivially, since it refers to a subset of the constraints for which the condition held at the beginning of the call. Using the inductive hypothesis, (Post1), (Post2), and the second set of constraints in (Post3) follow immediately. The only constraints that are left to be verified are
$$v[\sigma_1, \sigma_2] = \sum_{a' \in A_{I_2}} v[\sigma_1, (I_2, a')] \quad \forall I_2 \in \mathcal{I}_2 : \sigma \bowtie I_2, \sigma(I_2) = \sigma_2. \tag{3}$$

That constraint is guaranteed by Lines 7 and 8 for all $I_2 \neq I_1$. So, we need to verify that it holds for all those $I_2$ such that $\sigma(I_2) = \sigma_2, \sigma \bowtie I_2$ and $I_1 \rightleftharpoons I_2$. Let $I_2$ be one such information set. Then, from Lines 4 and 5 we have that
$$v[\sigma_1, (I_2, a)] = \sum_{a' \in A_{I_1}} v[(I, a'), (I_2, a)] \quad \forall a \in A_{I_2}.$$

Summing the above equations across all $a \in A_{I_2}$ and using (Pre7) yields
$$\begin{aligned}
\sum_{a \in A_{I_2}} v[\sigma_1, (I_2, a)] &= \sum_{a \in A_{I_2}} \sum_{a' \in A_{I_1}} v[(I, a'), (I_2, a)] \\
&= \sum_{a' \in A_{I_1}} \sum_{a \in A_{I_2}} v[(I, a'), (I_2, a)] \\
&= \sum_{a' \in A_{I_1}} v[(I, a'), \sigma(I_2)] \\
&= \sum_{a' \in A_{I_1}} v[(I, a'), \sigma_2],
\end{aligned}$$

where we used the hypothesis that $\sigma(I_2) = \sigma_2$ in the last equality. Finally, since $I_1 \rightleftharpoons I_2$ and $\sigma(I_2) = \sigma_2$, it must be $I_1 \bowtie \sigma_2$ and so, using (Pre6), we obtain that

$$\sum_{a \in A_{I_2}} v[\sigma_1, (I_2, a)] = v[\sigma_1, \sigma_2],$$

completing the proof of Equation (3). So, (Post3) holds as well and the proof of the inductive step is complete. $\qquad\square$

The inductive contract for FILLOUTCOLUMN is symmetric and we omit it.

### A.2.2 The Full Algorithm

---

**Algorithm 3:** DECOMPOSE$((\sigma_1, \sigma_2), \mathcal{S}, \mathcal{D})$

---

**Preconditions:** $(\sigma_1, \sigma_2) \in \Sigma_1 \bowtie \Sigma_2, (\sigma_1, \sigma_2) \in \mathcal{S}$

1   $B \leftarrow \emptyset$
2   **for** *all* $i \in \{1, 2\}, I \in \mathcal{I}_i, \sigma(I) = \sigma_i, \sigma_{-i} \bowtie I$ **do**
3     **if** *the* $\sigma_{-i}$-*rank of* $I$ *is* 0 **then**
4       $B \leftarrow B \sqcup I$
5   **for** $(I_1, I_2) \in \mathcal{I}_1 \times \mathcal{I}_2$ *such that* $\sigma(I_1) = \sigma_1, \sigma(I_2) = \sigma_2, I_1 \rightleftharpoons I_2$ **do**
6     **if** *the* $\sigma_2$-*rank of* $I_1$ *is* $\geq$ *the* $\sigma_1$-*rank of* $I_2$ **then**
7       $B \leftarrow B \sqcup I_1$
8     **else**
9       $B \leftarrow B \sqcup I_2$
10   **for** $I \in B$ **do**
11     **if** $I \in \mathcal{I}_1$ **then**
      ▷ Fill all $\{v[(I, a), \sigma_2] : a \in A_I\}$ by splitting $v[\sigma_1, \sigma_2]$ accordance with the von Stengel-Forges constraints
12       $\mathcal{S} \leftarrow \mathcal{S} \sqcup \{((I, a), \sigma_2) : a \in A_I\}$
13       $\mathcal{D} \leftarrow \mathcal{D} \triangleleft^h \Delta^{|A_I|}$ where $h : \boldsymbol{v} \mapsto v[\sigma_1, \sigma_2]$
      ▷ Recursive call
14       **for** $\sigma_1' \in \{(I, a) : a \in A_I\}$ **do**
15         DECOMPOSE$((\sigma_1', \sigma_2), \mathcal{S}, \mathcal{D})$
      ▷ Fill a portion of the row for $\sigma_1$
16       **for** $I_2 \in \mathcal{I}_2 : \sigma_1 \bowtie I_2, \sigma(I_2) = \sigma_2$ **do**
17         **for** $\sigma_2' \in \{(I_2, a') : a' \in A_{I_2}\}$ **do**
          ▷ Fill $(\sigma_1, \sigma_2')$ by summing up all entries $\{v[(I, a'), \sigma_2'] : a' \in A_I\}$ in accordance with the von Stengel-Forges constraints
18           $\mathcal{S} \leftarrow \mathcal{S} \sqcup \{(\sigma_1, \sigma_2')\}$
19           $\mathcal{D} \leftarrow \mathcal{D} \triangleleft^h \{1\}$ where $h : \boldsymbol{v} \mapsto \sum_{a' \in A_I} v[(I, a'), \sigma_2']$
20           FILLOUTROW$((\sigma_1, \sigma_2'), I)$
21     **else**
      ▷ Fill all $\{v[\sigma_1, (I, a)] : a \in A_I\}$ by splitting $v[\sigma_1, \sigma_2]$ accordance with the von Stengel-Forges constraints
22       $\mathcal{S} \leftarrow \mathcal{S} \sqcup \{(\sigma_1, (I, a)) : a \in A_I\}$
23       $\mathcal{D} \leftarrow \mathcal{D} \triangleleft^h \Delta^{|A_I|}$ where $h : \boldsymbol{v} \mapsto v[\sigma_1, \sigma_2]$
      ▷ Recursive call
24       **for** $\sigma_2' \in \{(I, a) : a \in A_I\}$ **do**
25         DECOMPOSE$((\sigma_1, \sigma_2'), \mathcal{S}, \mathcal{D})$
      ▷ Fill a portion of the column for $\sigma_2$
26       **for** $I_1 \in \mathcal{I}_1 : \sigma_2 \bowtie I_1, \sigma(I_1) = \sigma_1$ **do**
27         **for** $\sigma_1' \in \{(I_1, a') : a' \in A_{I_1}\}$ **do**
          ▷ Fill $(\sigma_1', \sigma_2)$ by summing up all entries $\{v[\sigma_1', (I, a')] : a' \in A_I\}$ in accordance with the von Stengel-Forges constraints
28           $\mathcal{S} \leftarrow \mathcal{S} \sqcup \{(\sigma_1', \sigma_2)\}$
29           $\mathcal{D} \leftarrow \mathcal{D} \triangleleft^h \{1\}$ where $h : \boldsymbol{v} \mapsto \sum_{a' \in A_I} v[\sigma_1', (I, a')]$
30           FILLOUTCOLUMN$((\sigma_1', \sigma_2), I)$
31   **return** $(\mathcal{S}, \mathcal{D})$

---

**Lemma 4** (Inductive contract for DECOMPOSE). *Assume that at the beginning of each call to* DECOMPOSE$((\sigma_1, \sigma_2), \mathcal{S}, \mathcal{D})$ *the following conditions hold*

*(Pre1)* $\mathcal{S}$ *contains only relevant sequence pairs and* $\mathcal{D}$ *consists of vectors indexed by exactly the indices in* $\mathcal{S}$.

*(Pre2)* $\mathcal{S}$ *does not contain any relevant sequence pairs which are descendants of* $(\sigma_1, \sigma_2)$, *with the only exception of* $(\sigma_1, \sigma_2)$ *itself. In formulas,*

$$\mathcal{S} \cap \{(\sigma_1', \sigma_2') \in \Sigma_1 \times \Sigma_2 : \sigma_1' \succeq \sigma_1, \sigma_2' \succeq \sigma_2\} = \{(\sigma_1, \sigma_2)\}.$$

*Then, at the end of the call, the returned sets* $(\mathcal{S}', \mathcal{D}')$ *are such that*

*(Post1)* $\mathcal{S}'$ *contains only relevant sequence pairs and* $\mathcal{D}'$ *consists of vectors* $v$ *indexed by exactly the indices in* $\mathcal{S}'$.

*(Post2)* *The call has filled in exactly all relevant sequence pair indices that are descendants of* $(\sigma_1, \sigma_2)$ *(except for* $(\sigma_1, \sigma_2)$ *itself, which was already filled in). In formulas,*

$$\mathcal{S}' = \mathcal{S} \sqcup \{(\sigma_1', \sigma_2') \in \Sigma_1 \times \Sigma_2 : \sigma_1' \succeq \sigma_1, \sigma_2' \succeq \sigma_2, (\sigma_1', \sigma_2') \neq (\sigma_1, \sigma_2), \sigma_1' \bowtie \sigma_2'\}.$$

*(Post3)* $\mathcal{D}'$ *satisfies the subset of von Stengel-Forges constraints*

$$\sum_{a \in A_I} v[(I, a), \sigma_2'] = v[\sigma(I), \sigma_2'] \quad \forall \sigma_2' \succeq \sigma_2, I \in \mathcal{I}_1 \text{ s.t. } \sigma_2' \bowtie I, \sigma(I) \succeq \sigma_1$$

$$\sum_{a \in A_J} v[\sigma_1', (J, a)] = v[\sigma_1', \sigma(J)] \quad \forall \sigma_1' \succeq \sigma_1, J \in \mathcal{I}_2 \text{ s.t. } \sigma_1' \bowtie J, \sigma(J) \succeq \sigma_2.$$

*Proof.* By induction.

- **Base case.** The base case is any $(\sigma_1, \sigma_2)$ such that there is no $\sigma_1' \succeq \sigma_1, \sigma_2' \succeq \sigma_2, \sigma_1' \bowtie \sigma_2'$. In that case, the set $B$ is empty, so the algorithm terminates immediately without modifying the sets $\mathcal{S}$ and $\mathcal{D}$. Consequently, (Post1) and (Post2) hold trivially from (Pre1) and (Pre2). (Post3) reduces to an empty set of constraints, so (Post3) holds as well.

- **Inductive step.** In order to use the inductive hypothesis, we will need to prove that the preconditions for DECOMPOSE hold on Lines 15 and 25. We will focus on Line 15 ($I \in \mathcal{I}_1$), as the analysis for the other case ($I \in \mathcal{I}_2$) is symmetric. (Pre1) clearly holds, since we always update $\mathcal{S}$ and $\mathcal{D}$ in tandem. Since all iterations of the **for** loop on Line 10 touch different information sets, at the time of the recursive call on Line 15, and given (Post2) for all previous recursive calls, the only relevant sequence pairs $(\sigma_1'', \sigma_2'')$ such that $\sigma_1'' \succeq \sigma_1', \sigma_2'' \succeq \sigma_2$ that have been filled are the ones on Lines 12 and 13. So, (Pre2) holds.
  We now check that the preconditions for FILLOUTROW hold at Line 20. (Pre1), (Pre2), (Pre3), and (Pre4) are trivial. (Pre5) and (Pre7) are guaranteed by (Post2) and (Post3) of DECOMPOSE applied to Line 15. (Pre6) holds because of Lines 18 and 19.
  Using the inductive contracts of FILLOUTROW, FILLOUTCOLUMN and DECOMPOSE for the recursive calls, we now show that all postconditions hold at the end of the call. (Post1) is trivial since we always update $\mathcal{S}$ and $\mathcal{D}$ together. (Post2) holds by keeping track of what entries are filled in Lines 12, 13, 18, 19, 22, 23, 28, 29, as well as those filled in the calls to FILLOUTROW, FILLOUTCOLUMN and DECOMPOSE, as regulated by postcondition (Post2) in the inductive contracts of the functions. In order to verify (Post3), we need to verify that the constraints that are not already guaranteed by the recursive calls hold. In particular, we need to verify that

  Ⓐ $\displaystyle\sum_{a \in A_I} v[(I, a), \sigma_2] = v[\sigma_1, \sigma_2] \quad \forall I \in \mathcal{I}_1 \text{ s.t. } \sigma_2 \bowtie I, \sigma(I) = \sigma_1, I \notin B$

  Ⓑ $\displaystyle\sum_{a \in A_J} v[\sigma_1, (J, a)] = v[\sigma_1, \sigma_2] \quad \forall J \in \mathcal{I}_2 \text{ s.t. } \sigma_1 \bowtie J, \sigma(J) = \sigma_2, J \notin B.$

  We will show that constraints Ⓐ hold; the proof for Ⓑ is symmetric. Using Lemma 2 together with the definition of $B$ (Lines 1-9), any information set $I \in \mathcal{I}_i : \sigma(I) = \sigma_i, \sigma_{-i} \bowtie I$ that is not in $B$ must have $\sigma_{-i}$-rank *exactly* 1. Let $I \in \mathcal{I}_1$ be such that $\sigma_2 \bowtie I, \sigma(I) = \sigma_1, I \notin B$, as required in Ⓐ. Since the $\sigma_2$-rank of $I$ is 1, let $J$ be the only information set in $\mathcal{I}_2$ such that $I \rightleftharpoons J, \sigma(J) = \sigma_2$. Note that $J \in B$. The entries $v[(I, a), \sigma_2] : a \in A_I$ were filled in Lines 28 and 29 when the **for** loop picked up $J \in B$. So, in particular,

  $$v[(I, a), \sigma_2] = \sum_{a' \in A_J} v[(I, a), (J, a')] \quad \forall a \in A_I.$$

Summing the above equations across $a \in A_I$, we obtain

$$\sum_{a \in A_I} v[(I, a), \sigma_2] = \sum_{a \in A_I} \sum_{a' \in A_J} v[(I, a), (J, a')]$$

$$= \sum_{a' \in A_J} \sum_{a \in A_I} v[(I, a), (J, a')]$$

$$= \sum_{a' \in A_J} v[\sigma_1, (J, a')]$$

$$= v[\sigma_1, \sigma_2],$$

where the last equation follows from the way the entries $v[\sigma_1, (J, a')] : a' \in A_J$ were filled in (Lines 22 and 23). This shows that the set of constraints Ⓐ hold. ☐

**Theorem 2.** *The von Stengel-Forges polytope $\mathcal{V}$ of a two-player perfect-recall triangle-free EFG can be expressed via a sequence of scaled extensions with simplexes and singleton sets:*

$$\mathcal{V} = \{1\} \overset{h_1}{\triangleleft} \mathcal{X}_1 \overset{h_2}{\triangleleft} \mathcal{X}_2 \overset{h_3}{\triangleleft} \cdots \overset{h_n}{\triangleleft} \mathcal{X}_n, \tag{2}$$

*where, for $i = 1, \ldots, n$, either $\mathcal{X}_i = \Delta^{s_i}$ for some simplex dimension $s_i \in \mathbb{N}$, or $\mathcal{X}_i = \{1\}$, and $h_i$ is a linear function. Furthermore, an exact algorithm exists to compute such expression in linear time in the dimensionality of $\mathcal{V}$, and so, in time at most quadratic in the size of the game.*

*Proof.* The correctness of the algorithm follow from (Post3) in the inductive contract. Every time the set of partially-filled-in vectors $\mathcal{D}$ gets extended, it is extended with either the singleton set $\{1\}$ or a simplex. In either case the nonnegative affine functions $h$ used are linear. So, the decomposition structure is as in the statement. Finally, since the overhead of each call (on top of the recursive calls) is linear in the number of relevant sequence pairs $(\sigma, \tau) \in \Sigma_1 \bowtie \Sigma_2$ that are filled, and each relevant sequence pair is filled only once, the complexity of the algorithm is linear in the number of relevant sequence pairs. ☐

# B Relationship Between $\mathcal{V}$ and $\Xi$

## B.1 Preliminaries: Definition of the Polytope of Correlation Plans

Let $\Pi_i(\sigma)$ denote the subset of reduced-normal-form plans $\Pi_i$ for Player $i$ prescribe all actions of Player $i$ on the path from the root of the game down to the information set-action pair $\sigma$ (if $\sigma =$, assign $\Pi_i(\varnothing) = \Pi_i$). The transformation from a correlated distribution $\mu$ to its correlation plan representation is achieved using a linear function

$$f : \Delta^{|\Pi_1 \times \Pi_2|} \to \mathbb{R}_{\geq 0}^{|\Sigma_1 \bowtie \Sigma_2|}.$$

Specifically, $f$ takes a generic distribution $\mu$ over $\Pi_1 \times \Pi_2$ and maps to the vector $\boldsymbol{\xi} = f(\mu)$, called a *correlation plan*, whose components are

$$\xi[\sigma_1, \sigma_2] := \sum_{\pi_1 \in \Pi_1(\sigma_1)} \sum_{\pi_2 \in \Pi_2(\sigma_2)} \mu(\pi_1, \pi_2) \qquad \forall (\sigma_1, \sigma_2) \in \Sigma_1 \bowtie \Sigma_2. \tag{4}$$

The set of all valid correlation plans, $\Xi$, is defined as the image $\mathrm{Im}\, f$ of $f$ as the distribution $\mu$ takes any possible value in $\Delta^{|\Pi_1 \times \Pi_2|}$.

**Remark 2.** *Since $f$ sums up distinct entries from the distribution $\mu$, all entries in $\boldsymbol{\xi} = f(\mu)$ are in the range $[0, 1]$.*

## B.2 Proofs

**Lemma 5.** *Let $\mathbf{1}_{(\pi_1, \pi_2)} \in \Delta^{|\Pi_1 \times \Pi_2|}$ denote the distribution over $\Pi_1 \times \Pi_2$ that assigns mass $1$ to the pair $(\pi_1, \pi_2)$, and mass $0$ to any other pair of reduced-normal-form plans. Then,*

$$\Xi = \mathrm{co}\{f(\mathbf{1}_{(\pi_1, \pi_2)}) : \pi_1 \in \Pi_1, \pi_2 \in \Pi_2\}.$$

*Proof.* The "deterministic" distributions $\mathbf{1}_{(\pi_1, \pi_2)}$ are the vertices of $\Delta^{|\Pi_1 \times \Pi_2|}$, so, in particular,

$$\Delta^{|\Pi_1 \times \Pi_2|} = \mathrm{co}\{\mathbf{1}_{(\pi_1, \pi_2)} : \pi_1 \in \Pi_1, \pi_2 \in \Pi_2\}.$$

Since by definition $\Xi = \mathrm{Im}\, f$, and $f$ is a linear function, the images (under $f$) of the $\mathbf{1}_{(\pi_1, \pi_2)}$ are a convex basis for $\Xi$, which is exactly the statement. ☐

**Lemma 6.** *Let $v \in \mathcal{V}$. For all $\sigma_1 \in \Sigma_1$ such that $v[\sigma_1, \varnothing] = 0$, $v[\sigma_1, \sigma_2] = 0$ for all $\sigma_2 \bowtie \sigma_1$. Similarly, for all $\sigma_2 \in \Sigma_2$ such that $v[\varnothing, \sigma_2] = 0$, $v[\sigma_1, \sigma_2] = 0$ for all $\sigma_1 \bowtie \sigma_2$.*

*Proof.* We prove the theorem by induction on the *depth* of the sequences $\sigma_1$ and $\sigma_2$. The depth $\mathrm{depth}(\sigma)$ of a generic sequence $\sigma = (I, a) \in \Sigma_i$ of Player $i$ is defined as the number of actions that Player $i$ plays on the path from the root of the tree down to action $a$ at information set $I$ included. Conventionally, we let the depth of the empty sequence be 0.

Take $\sigma_1 \in \Sigma_1$ such that $v[\sigma_1, \varnothing] = 0$. For $\sigma_2$ of depth 0 (that is, $\sigma_2 = \varnothing$), clearly $v[\sigma_1, \sigma_2] = 0$. For the inductive step, suppose that $v[\sigma_1, \sigma_2] = 0$ for all $\sigma_2 \in \Sigma_2, \sigma_1 \bowtie \sigma_2$ such that $\mathrm{depth}(\sigma_2) \leq d_2$. We will show that $v[\sigma_2, \sigma_2] = 0$ for $\mathrm{depth}(\sigma_2) \leq d_2 + 1$. Indeed, let $(I, a') = \sigma_2 \bowtie \sigma_1$ of depth $d_2 + 1$. Since $v \in \mathcal{V}$, in particular the von Stengel-Forges constraint $\sum_{a \in A_I} v[\sigma_1, (I, a)] = v[\sigma_1, \sigma(I)]$ must hold. The depth of $\sigma(I)$ is $d_2$, so by the inductive hypothesis, it must be $v[\sigma_1, \sigma(I)] = 0$, and therefore $\sum_{a \in A_I} v[\sigma_1, (I, a)] = 0$. But all entries of $v$ are nonnegative, so it must be $v[\sigma_1, (I, a)] = 0$ for all $a \in A_I$, and in particular for $(I, a') = \sigma_2$. This completes the proof by induction.

The proof for the second part is analogous. $\qquad\qquad\square$

**Lemma 7.** *Let $v \in \mathcal{V}$ have integer $\{0, 1\}$ coordinates. Then, for all $(\sigma_1, \sigma_2) \in \Sigma_1 \bowtie \Sigma_2$, it holds that*

$$v[\sigma_1, \sigma_2] = v[\sigma_1, \varnothing] \cdot v[\varnothing, \sigma_2].$$

*Proof.* We prove the theorem by induction on the depth of the sequences, similarly to Lemma 6.

The base case for the induction proof corresponds to the case where $\sigma_1$ and $\sigma_2$ both have depth 0, that is, $\sigma_1 = \sigma_2 = \varnothing$. In that case, the theorem is clearly true, because $v[\varnothing, \varnothing] = 1$ as part of the von Stengel-Forges constraints (1).

Now, suppose that the statement holds as long as $\mathrm{depth}(\sigma_1), \mathrm{depth}(\sigma_2) \leq d$. We will show that the statement will hold for any $(\sigma_1, \sigma_2) \in \Sigma_1 \bowtie \Sigma_2$ such that $\mathrm{depth}(\sigma_1), \mathrm{depth}(\sigma_2) \leq d + 1$. Indeed, consider $(\sigma_1, \sigma_2) \in \Sigma_1 \bowtie \Sigma_2$ such that $\mathrm{depth}(\sigma_1), \mathrm{depth}(\sigma_2) \leq d + 1$. If any of the sequences is the empty sequence, the statements holds trivially, so assume that neither is the empty sequence and in particular $\sigma_1 = (I, a), \sigma_2 = (J, b)$. If $v[\sigma_1, \varnothing] = 0$, then from Lemma 6 $v[\sigma_1, \sigma_2] = 0$ and the statement holds. Similarly, if $v[\varnothing, \sigma_2] = 0$, then $v[\sigma_1, \sigma_2] = 0$, and the statement holds. Hence, the only remaining case given the integrality assumption on the coordinates of $v$ is $v[\sigma_1, \varnothing] = v[\varnothing, \sigma_2] = 1$.

From the von Stengel-Forges constraints, $v[\sigma(I), \varnothing] = \sum_{a' \in A_I} v[(I, a'), \varnothing] = 1 + \sum_{a' \in A_I, a' \neq a} v[(I, a'), \varnothing] \geq 1$. Hence, because all entries of $v$ are in $\{0, 1\}$, it must be $v[\sigma(I), \varnothing] = 1$ and $v[(I, a'), \varnothing] = 0$ for all $a' \in A_I, a' \neq a$. With a similar argument we conclude that $v[\varnothing, \sigma(J)] = 1$ and $v[\varnothing, (J, b')] = 0$ for all $b' \in A_J, b \neq b'$. Using the inductive hypothesis, $v[\sigma(I), \sigma(J)] = v[\sigma(I), \varnothing] \cdot v[\varnothing, \sigma(J)] = 1$.

Now, using the von Stengel-Forges constraints together with the equality $v[\sigma(I), \sigma(J)] = 1$ we just proved, we conclude that

$$\sum_{a' \in A_I} \sum_{b' \in A_J} v[(I, a'), (J, b')] = 1. \tag{5}$$

On the other hand, since $v[(I, a'), \varnothing] = 0$ for all $a' \in A_I, a' \neq a$ and $v[\varnothing, (J, b')] = 0$ for all $b' \in A_J, b' \neq b$, from Lemma 6 we have that

$$a' \neq a \lor b' \neq b \implies v[(I, a'), (J, b')] = 0. \tag{6}$$

From (6) and (5), we conclude that $v[(I, a), (J, b)] = v[\sigma_1, \sigma_2] = 1 = v[\sigma_1, \varnothing] \cdot v[\varnothing, \sigma_2]$, as we wanted to show. $\qquad\square$

**Theorem 3.** *Let $\Gamma$ be a two-player perfect-recall extensive-form game, let $\mathcal{V}$ be its von Stengel-Forges polytope, and let $\Xi$ be its polytope of correlation plans. Then, $\Xi = \mathcal{V}$ if and only if all vertices of $\mathcal{V}$ have integer $\{0, 1\}$ coordinates.*

*Proof.* We prove the two implications separately.

($\Rightarrow$) We start by proving that if $\Xi = \mathcal{V}$, then all vertices of $\mathcal{V}$ have integer $\{0, 1\}$ coordinates. Since $\mathcal{V} = \Xi$ by hypothesis, from 5 we can write

$$\mathcal{V} = \mathrm{co}\{f(\mathbf{1}_{(\pi_1, \pi_2)}) : \pi_1 \in \Pi_1, \pi_2 \in \Pi_2\}.$$

So, to prove this direction it is enough to show that $f(\mathbf{1}_{(\pi_1,\pi_2)})$ has integer $\{0,1\}$ coordinates for all $(\pi_1,\pi_2) \in \Pi_1 \times \Pi_2$. To see that, we use the definition (4): each entry in $f(\mathbf{1}_{(\pi_1,\pi_2)})$ is the sum of distinct entries of $\mathbf{1}_{(\pi_1,\pi_2)}$. Given that by definition $\mathbf{1}_{(\pi_1,\pi_2)}$ has exactly one entry with value 1 and $|\Pi_1 \times \Pi_2| - 1$ entries with value 0, we conclude that all coordinates of $f(\mathbf{1}_{(\pi_1,\pi_2)})$ are in $\{0,1\}$.

($\Leftarrow$) We now show that if all vertices of $\mathcal{V}$ have integer $\{0,1\}$ coordinates, then $\mathcal{V} \subseteq \Xi$. This is enough, since the reverse inclusion, $\mathcal{V} \supseteq \Xi$, is trivial and already known [26]. Let $\{\mathbf{v}_1, \ldots, \mathbf{v}_n\}$ be the vertices of $\mathcal{V}$. To conclude that $\mathcal{V} \subseteq \Xi$, we will prove that $\mathbf{v}_i \in \Xi$ for all $i = 1, \ldots, n$. This will be sufficient since both $\mathcal{V}$ and $\Xi$ are convex.

Let $\mathbf{v} \in \{\mathbf{v}_1, \ldots, \mathbf{v}_n\}$ be any vertex of $\mathcal{V}$. By hypothesis, $v[\sigma_1, \sigma_2] \in \{0,1\}$ for all $(\sigma_1, \sigma_2) \in \Sigma_1 \bowtie \Sigma_2$. Because $\mathbf{v}$ satisfies the von Stengel-Forges constraints and furthermore $\mathbf{v}$ has $\{0,1\}$ entries by hypothesis, the two vectors $\mathbf{q}_1, \mathbf{q}_2$ defined according to $\mathbf{q}_1[\sigma_1] = v[\sigma_1, \varnothing]$ $(\sigma_1 \in \Sigma_1)$ and $\mathbf{q}_2[\sigma_2] = v[\varnothing, \sigma_2]$ $(\sigma_2 \in \Sigma_2)$ are *pure* sequence-form strategies. Now, let $\pi_1^*$ and $\pi_2^*$ be the reduced-normal form plans corresponding to $\mathbf{q}_1$ and $\mathbf{q}_2$, respectively. We will show that $\mathbf{v} = f(\mathbf{1}_{(\pi_1^*,\pi_2^*)})$, which will immediately imply that $\mathbf{v} \in \Xi$ using Lemma 5.

Since $\mathbf{1}_{(\pi_1^*,\pi_2^*)}$ has exactly one positive entry with value 1 in the position corresponding to $(\pi_1^*,\pi_2^*)$, by definition of the linear map $f$, for any $(\sigma_1, \sigma_2) \in \Sigma_1 \bowtie \Sigma_2$,

$$f(\mathbf{1}_{(\pi_1^*,\pi_2^*)})[\sigma_1,\sigma_2] = \mathbb{1}[\sigma_1 \in \Pi_1(\sigma_1)] \cdot \mathbb{1}[\sigma_2 \in \Pi_2(\sigma_2)]. \tag{7}$$

So, using the known properties of pure sequence-form strategies, we obtain

$$f(\mathbf{1}_{(\pi_1^*,\pi_2^*)})[\sigma_1,\sigma_2] = q_1[\sigma_1] \cdot q_2[\sigma_2] = v[\sigma_1, \varnothing] \cdot v[\varnothing, \sigma_2] = v[\sigma_1,\sigma_2],$$

where the last equality follows from Lemma 7. Since the equality holds for any $(\sigma_1, \sigma_2) \in \Sigma_1 \bowtie \Sigma_2$, we have that $\mathbf{v} = f(\mathbf{1}_{(\pi_1^*,\pi_2^*)})$. $\qquad \square$

**Lemma 1.** *Let $\mathcal{X}, \mathcal{Y}$, and $h$ be as in Definition 1. If $\mathcal{X}$ is a convex polytope with vertices $\{\mathbf{x}_1, \ldots, \mathbf{x}_n\}$, and $\mathcal{Y}$ is a convex polytope with vertices $\{\mathbf{y}_1, \ldots, \mathbf{y}_m\}$, then $\mathcal{X} \triangleleft^h \mathcal{Y}$ is a convex polytope whose vertices are a nonempty subset of $\{(\mathbf{x}_i, h(\mathbf{x}_i)\mathbf{y}_j) : i \in \{1, \ldots, n\}, j \in \{1, \ldots, m\}\}$.*

*Proof.* Take any point $\mathbf{z} \in \mathcal{X} \triangleleft^h \mathcal{Y}$. By definition of scaled, extension, there exist $\mathbf{x} \in \mathcal{X}, \mathbf{y} \in \mathcal{Y}$ such that $\mathbf{z} = (\mathbf{x}, h(\mathbf{x})\mathbf{y})$. Since $\{\mathbf{x}_1, \ldots, \mathbf{x}_n\}$ are the vertices of $\mathcal{X}$, $\mathbf{x}$ can be written as a convex combination $\mathbf{x} = \sum_{i=1}^{n} \lambda_i \mathbf{x}_i$ where $(\lambda_1, \ldots, \lambda_n) \in \Delta^n$. Similarly, $\mathbf{y} = \sum_{i=1}^{m} \mu_i \mathbf{y}_i$ for some $(\mu_1, \ldots, \mu_m) \in \Delta^m$. Hence, using the hypothesis that $h$ is affine, we can write

$$\mathbf{z} = (\mathbf{x}, h(\mathbf{x})\mathbf{y})$$

$$= \left( \sum_{i=1}^{n} \lambda_i \mathbf{x}_i, h\left( \sum_{i=1}^{n} \lambda_i \mathbf{x}_i \right) \sum_{j=1}^{m} \mu_j \mathbf{y}_j \right)$$

$$= \left( \sum_{i=1}^{n} \lambda_i \mathbf{x}_i, \left( \sum_{i=1}^{n} \lambda_i h(\mathbf{x}_i) \right) \sum_{j=1}^{m} \mu_j \mathbf{y}_j \right)$$

$$= \sum_{i=1}^{n} \sum_{j=1}^{m} \lambda_i \mu_j (\mathbf{x}_i, h(\mathbf{x}_i)\mathbf{y}_j).$$

Since $\lambda_i \mu_j \geq 0$ for all $i \in \{1, \ldots, n\}, j \in \{1, \ldots, m\}$ and $\sum_{i=1}^{n} \sum_{j=1}^{m} \lambda_i \mu_j = (\sum_{i=1}^{n} \lambda_i)(\sum_{j=1}^{m} \mu_j) = 1$, we conclude that $\mathbf{z} \in \mathrm{co}\{(\mathbf{x}_i, h(\mathbf{x}_i)\mathbf{y}_j) : i \in \{1, \ldots, n\}, j \in \{1, \ldots, m\}\}$. On the other hand, $(\mathbf{x}_i, h(\mathbf{x}_i)\mathbf{y}_j) \in \mathcal{X} \triangleleft^h \mathcal{Y}$, so

$$\mathcal{X} \overset{h}{\triangleleft} \mathcal{Y} = \mathrm{co}\{(\mathbf{x}_i, h(\mathbf{x}_i)\mathbf{y}_j) : i \in \{1, \ldots, n\}, j \in \{1, \ldots, m\}\}.$$

Since the vertices of a (nonempty) polytope are a (nonempty) subset of any convex basis for the polytope, the vertices of $\mathcal{X} \triangleleft^h \mathcal{Y}$ must be a nonempty subset of $\{(\mathbf{x}_i, h(\mathbf{x}_i)\mathbf{y}_j) : i \in \{1, \ldots, n\}, j \in \{1, \ldots, m\}\}$, which is the statement. $\qquad \square$

**Theorem 4.** *Let $\mathcal{V}$ be the von Stengel-Forges polytope of a two-player triangle-free game (Definition 3). All vertices of $\mathcal{V}$ have integer $\{0,1\}$ coordinates.*

*Proof.* We prove the statement by induction over the scaled-extension-based decomposition

$$\mathcal{V} = \{1\} \overset{h_1}{\triangleleft} \mathcal{X}_1 \overset{h_2}{\triangleleft} \cdots \overset{h_n}{\triangleleft} \mathcal{X}_n.$$

In particular, we will show that for all $k = 0, \ldots, n$, the coordinates of the vertices of the polytope

$$\mathcal{V}_k = \{1\} \overset{h_1}{\triangleleft} \cdots \overset{h_k}{\triangleleft} \mathcal{X}_k$$

constructed by considering only the first $k$ scaled extensions in the decomposition are all integer. Since $\mathcal{V} \subseteq [0,1]^{|\Sigma_1 \bowtie \Sigma_2|}$ (Remark 2), this immediately implies that each coordinate is in $\{0,1\}$.

- **Base case:** $k = 0$. In this case, $\mathcal{V}_0 = \{1\}$. The only vertex is $\{1\}$, which is integer. So, base case trivially holds.

- **Inductive step.** Suppose that the polytope $\mathcal{V}_k$ $(k < n)$ has integer vertices. We will show that the same holds for $\mathcal{V}_{k+1}$. Clearly, $\mathcal{V}_{k+1} = \mathcal{V}_k \triangleleft^{h_{k+1}} \mathcal{X}_{k+1}$. From the properties of the structural decomposition, we know that $\mathcal{K}_{k+1}$ is either the singleton $\{1\}$, or a probability simplex $\Delta^{s_{k+1}}$ for some appropriate dimension $s_{k+1}$. We break the analysis accordingly.

  - If $\mathcal{X}_{k+1} = \{1\}$, the scaled extension represents filling in a linearly-dependent entry in $\boldsymbol{v} \in \mathcal{V}$ by summing already-filled-in entries. So, $h_{k+1}$ takes a partially-filled-in vector from $\mathcal{V}_k$ and sums up some of its coordinates. Let $\boldsymbol{v}_1, \ldots, \boldsymbol{v}_n$ be the vertices of $\mathcal{V}_k$. Using Lemma 1, the vertices of $\mathcal{V}_{k+1}$ are a subset of

    $$\{(\boldsymbol{v}_i, h(\boldsymbol{v}_i) \cdot 1) : i = 1, \ldots, n\}. \tag{8}$$

    Since by inductive hypothesis $\boldsymbol{v}_i$ have integer coordinates, and $h$ sums up some of them, $h(\boldsymbol{v})_i$ is integer for all $i = 1, \ldots, n$. So, all of the vectors in (8) have integer coordinates, and in particular this must be true of the vertices of $\mathcal{V}_{k+1}$.

  - If $\mathcal{X}_{k+1} = \Delta^{s_{k+1}}$, the scaled extension represents the operation of partitioning an already-filled-in entry $v[\sigma, \tau]$ of $\mathcal{V}_k$ into $s_i$ non-negative real values. The affine function $h_{k+1}$ extracts the entry $v[\sigma, \tau]$ from each vector $\boldsymbol{v} \in \mathcal{V}_k$. Let $\boldsymbol{v}_1, \ldots, \boldsymbol{v}_n$ be the vertices of $\mathcal{V}_k$. The vertices of $\Delta^{s_{k+1}}$ are the canonical basis vectors $\{\boldsymbol{e}_1, \ldots, \boldsymbol{e}_{s_{k+1}}\}$. From Lemma 1, the vertices of $\mathcal{V}_{k+1}$ are a subset of

    $$\{(\boldsymbol{v}_i, h(\boldsymbol{v}_i)\boldsymbol{e}_j) : i = 1, \ldots, n, j = 1, \ldots, s_{k+1}\}$$
    $$= \{(\boldsymbol{v}_i, v_i[\sigma, \tau]\boldsymbol{e}_j) : i = 1, \ldots, n, j = 1, \ldots, s_{k+1}\}. \tag{9}$$

    Since by inductive hypothesis the vertices $\boldsymbol{v}_i$ have integer coordinates, $v_i[\sigma, \tau]$ is an integer. Since the canonical basis vector only have entries in $\{0,1\}$, all of the vectors in (9) have integer coordinates. So, in particular, this must be true of the vertices of $\mathcal{V}_{k+1}$. □

## C  Additional Experimental Results

In this section we present additional computational results. Specifically, we present results on how well algorithms can solve for EFCE (and thus also EFCCE and NFCCE since they are supsets of EFCE) after our new scaled-extension-based structural decomposition has been computed for the polytope of correlation plans using the algorithm that we presented in the body. The speed of that algorithm for computing the decomposition is extremely fast, as shown in the body both theoretically and experimentally. Here we report the performance of two leading algorithms for finding an approximate optimal EFCE after the decomposition algorithm has completed. Specifically, we compare the performance of the regret-minimization method of Farina et al. [14] to that of the barrier algorithm for linear programming implemented by the Gurobi commercial linear programming solver, as described in the body of the paper. (On these problems, any linear programming solver could be used in principle, but simplex and dual simplex methods—even the ones in Gurobi—are prohibitively slow. Similarly, the subgradient descent method of Farina et al. [13] is known to be dominated by the regret-minimization method of Farina et al. [14].)

Both algorithms are used to converge to a feasible EFCE—that is, no objective function was set—in the largest Goofspiel instance ($k = 5$). Our implementation of the regret minimization method is single-threaded, while we allow Gurobi to use 30 threads. All experiments were conducted on a machine with 64 cores and 500GB of memory. Gurobi required roughly 200GB of memory, while the memory footprint of the regret-minimization algorithm was less than 2GB.

At all times, the regret-minimization algorithm produces *feasible* correlation plans, that is, points that belong to $\Xi = \mathcal{V}$. So, that algorithm's iterates' infeasibility is defined as how incentive-incompatible the computed correlation plan is, measured as the difference in value that each player would gain by optimally deviating from any recommendation at any information set in the game. In contrast, the barrier method does not guarantee that the correlation plan is primal feasible, that is, the correlation

plans produced by the barrier algorithm might not be in $\Xi = \mathcal{V}$. Therefore, for Gurobi, we measure infeasibility as the maximum between (i) the (maximum) violation of the constraints that define $\mathcal{V}$, and (ii) the incentive-incompatibility of the iterate.

Figure 4 shows the results. The regret minimization algorithm works better as an anytime algorithm and leads to lower infeasibility for most of the run. The barrier method needs significant time to preprocess before even the first iterates are found. After that it converges rapidly.

Figure 4: Performance of the regret minimization method of Farina et al. [14] compared to Gurobi's barrier method in the largest Goofspiel game ($k = 5$).