[Reviews · NeurIPS 2020]

Review 1

Summary and Contributions: The paper characterizes a new, larger, subset of games where an optimal correlated equilibrium can be computed in polynomial time, using the concept of triangle-free games. The polytope of von Stengel and Forge, V, can be decomposed and an algorithm built on that decomposition. In the case of triangle-free games, V is shown to be equivalent to the set of correlation plans.

Strengths: The set of triangle-free games is a non-trivial extension of the class of extensive-form games where it is known to be possible to compute an optimal correlated equilibrium: it now includes games with some types of chance events - even including some (variants of?) real world games.

Weaknesses: While there was nothing in particular wrong with the experiments, I wasn't sure what I was supposed to take away from the experiments. Maybe also have some explanation of how decomposing the polytope now allows it to be mapped?

Correctness: Yes, with the caveats mentioned above w.r.t. experiments.(maybe some explanation of how decomposing the polytope now allows it to be mapped?)

Clarity: The paper is generally well written, but I found myself getting confused between whether an actual value is being assigned to a policy. For example, in the First Example, "Then, this value is split arbitrarily into the two (non-negative) entries v[∅, 1], v[∅, 2] so that v[∅, 1] + v[∅It was not impossible to read, but , 2] = v[∅, ∅]" constructing one specific policy, while the describing the (general version of the) operation as the step in constructing the polytope. The later usage of terms like "filled in" in the algorithm and its description are then harder to immediately parse: as far as I can tell values for sequences are not actually being filled in, but instead we are marking down that a particular constaint about filling in values has now been added to the scaled-extension polytope. I found the boxes in Figure 2 slightly confusing as well. Following the text for the leftmost example, going from v[0,1] to v[1,1] + v[2,1] and v[3,1] + v[4,1], I would have expected to see a longer column in the figure for those fields. (Or conversely, given the figure, I would have expected to go from v[0,1] and v[0,2] to v[1,1]+v[2,1] and v[1,2]+v[2,2].) The connection between the boxes and what is going on should be made more clear. Consider using some of the space from the experiment section to expand section 3. For example, a sketch of the argument for Theorem 3 seems more informative than knowing that Gurobi struggles to solve the LP based on the decomposition (when this does not really seem to be part of message of this paper.)

Relation to Prior Work: Yes, the distinction is clear, and appears to cite relevant related works.

Reproducibility: Yes

Additional Feedback: line 244 "topmost game in Figure 2" leftmost ---- comments after feedback Thank you for the response. It does clarify what the experimental section is trying to show, and I think making those points explicit would also help the paper. a) the proof says you can safely run an LP based on V rather than Xi -- which you could have done before, just without a theoretical argument of soundness. b) one could actually implement and run the decomposition algorithm that is part of the proof, whichis important because of (c) c) the regret-based alogrithm, which *does* require the decomposition, is (potentially much) faster than the LP algorithm.


Review 2

Summary and Contributions: The paper builds on work of on tractable computation of EFCE in two player games. Prior art permitted (tractable) solving two player extensive form games without chance moves. This work extends this capability to games with public chance moves - and more generally to two player EFGs that satisfy a "triangle-freeness" condition. The authors provides proofs, algorithms, and empirical validation of their methods.

Strengths: The paper story is clear and well motivated, and represents an exciting complexity result in EFGs. The figures are beautiful. Empirical evaluation is limited but perhaps sufficient to support the result. I think this is relevant (if a bit niche) to the NeurIPS community, particularly as a lot of the prior work was published in NuerIPS. All in all, a good submission - I enjoyed reading it and learned a thing or two!

Weaknesses: I found the algorithm a little intimidating. Although this cannot always be helped - I feel there could be some clarity improvements. I think the algorithm as presented in the appendix in long form was clearer. I have some nitpicks about the broader impact section (I expand on this in additional feedback). I also think some modest expansion of the empirical results would be of benefit to some readers (again I expand on this in additional feedback).

Correctness: Theoretical claims appear to be correct. Empirical results are plausible. Appendices were not checked for correctness.

Clarity: The technical contributions in this paper are complex. While I feel the authors to try to lead the reader in gently, there is a wall of notation and algorithmic complexity to digest that is daunting. I found the examples and accompanying figure very helpful, and the triangle-freeness easy to understand with the aid of these examples. The algorithm in the main paper could be formatted more clearly. I think the appendix does a better job here. Some undefined values appear (lower case xi) and annoyingly there are differences in notation between main paper and appendix (eg xi to v). I feel that the more complex the arguments a paper makes, the more effort should be expended in optimizing for clarity, otherwise it risks intimidating readers. As it stands, the paper is fairly intimidating and there is room for improvement in terms of clarity. Low hanging fruit: algorithm formatting, pedantic defining of variables, consistency of variables between main paper and appendix. At a high level, I found the story and motivation to be clear, and understand how it fits into the bigger picture.

Relation to Prior Work: Prior work and novel contributions are clearly marked. The paper claims boldly that this is "the biggest positive complexity result surrounding extensive-form correlation in more than a decade" - I have ran this by my colleagues and believe this claim to be true.

Reproducibility: Yes

Additional Feedback: I can see why the authors would call the condition "triangle free", but favoring explicitness, can the authors add a sentence or two giving intuition for the name? Decompose algorithm: can this be presented more clearly? If the loops are nested, can they be indented? I found the phrase "we branch on Player 1" / branching confusing. I don't think the cited paper [12] uses the term "branch" in it either. I also find the sudden use of lower-case Xi jarring - is this a typo? (ninja edit: algorithms in appendix are clearer). Line 347: can the authors also recommend/cite an open source or free LP solver that might also be able to scale to similar sized games to aid reproducibility? Broader impact: Although I think the authors have struck a good balance in this section, and I agree broadly with the potential positive/negative impacts that the authors suggest, I take slight issue with some of the phrasing. Firstly, it is implies that CEs do not have an equilibrium selection problem "to select, among the infinite number of correlated equilibria of the game, *one* that maximizes a given objective". There is still an equilibrium selection problem in constant-sum games, or general-sum games with a linear objective, and, for example, in the example in Figure 3 shows this where there is a frontier which can trade off payoff between the players (the authors do later mention this in the last sentence). Secondly, I dislike using the term "maximize social welfare", (from "For example, this technology could be used to find correlated equilibria than maximize social welfare, leading to highest societal good".). Although I recognize that "maximum social welfare" is a technical term used in the literature, I feel it is a misnomer, particularly in the context of a section discussing potential ethical impact. For example, even if we agreed that equilibria that provide "highest societal good" satisfy linear objective functions (something I do not believe), all the solutions on the frontier in Figure 3 are technically "maximum social welfare" solutions as defined by the literature - but I don't think you will find many people agreeing that all those equilibria are equally "leading to highest societal good". Can lower k-value Goofspiel Performance results also be added please? Although I understand that the point of the figure is to show that the problem is now tractable - for others seeking to reimplement the idea, having a baseline to check against that doesn't require 200GB of RAM and a commercial LP license would be helpful. On a similar line of argument, could the same plots vs iteration count be provided because wall-clock is hard to compare between machines when attempting to reproduce results. I am a little confused about the scales of the infeasibility. Am I right in thinking that there are 2.3e7 actions to spread mass over? And therefore if mass was spread uniformly each would get 1/2.3e7? And the maximum possible payoff is not more than 5? Why do the infeasibility constraints start around 10e2? And is 10e-8 only a little bit better than what one would obtain with a uniform distribution (to a first approximation)? Has Goofspiel been (CE) solved before (if so, please cite)? Can your approach solve it to an acceptable tolerance? If so, is this worth highlighting as a contribution? *** Thanks for your replies. I hadn't appreciated that the LP could be run (unsoundly) before, which is why I am reducing my score by one point. However I think there is a contribution here and think that extending [12] to public chance / triangle free games is a non-trivial result.


Review 3

Summary and Contributions: This paper studies the complexity of computing an optimal correlated equilibrium in two-player extensive-form games. It has been only known that an optimal correlated equilibrium in games without chance nodes can be computed in polynomial time. This paper shows that an optimal correlated equilibrium in triangle-free games, including games with public chance nodes, can be computed in polynomial time. 

Strengths: This paper shows that an optimal correlated equilibrium in triangle-free games can be computed in polynomial time. To achieve this, this paper shows that the set of correlation plans of a triangle-free game coincides with the von Stengel-Forges polytope of the game based on a structural decomposition, where the polytope that only requires a polynomial number of linear “probability-mass-conserving” constraints. Sound theoretical results are provided.  

Weaknesses: The novelty of this paper is very limited. The approach in this paper follows the existing work (Farina et al. [12] ) for games without chance nodes, and the techniques in both papers are almost the same except that this paper addresses specific features in triangle-freeness games by extending the techniques in Farina et al. [12]. Unfortunately, this extension is very straightforward. Actually, the key steps leading to the results in this paper are almost the same as the work in Farina et al. [12]. For example, Remark 1 in both papers raise the same problem for the decomposition algorithm; The definition of a triangle-freeness game in Definition 3 in this paper is almost the property of Proposition 2 in Farina et al. [12] except for simply adding I_1 \rightleftharpoons J_2. And the decomposition algorithm follows the one in Farina et al. [12] by handing the specific feature in the branching step. Then, it seems that this paper just tells us that the work in Farina et al. [12] is straightforward to extend to the triangle-freeness game.  Triangle-freeness games are very special games. It is not clear whether there are many triangle-freeness games in the real world. *********************** Thanks for your response. I have read the author's response and my opinion remains the same because I still cannot see why the extension is not easy.

Correctness: Seems correct.

Clarity: Satisfied.

Relation to Prior Work: Clear.

Reproducibility: Yes

Additional Feedback: This paper highlights that triangle-freeness games are far more general games than games without chance moves, which needs more explanations.    It will be better if illustrating how to avoid the problem in the triangle-freeness game. In Line 244, ‘the topmost game in Figure 2’ is not clear as three games are in the same row.  In Line 349, the sentence ‘However, even that quickly becomes impractical’ is incomplete. What is ‘infeasibility’? A formal definition is better.


Review 4

Summary and Contributions: In this paper, the authors study the problem of computing several formulations of correlated equilibrium with an objective function in two-player sequential games in polynomial time. It is well known that these solution concepts can be computed efficiently in case the game does not contain random events, but the problem remains intractable if such events occur. The main result of the authors further refines this boundary by showing that if a game satisfies a so-called triangle-freeness condition, the solution can be found in polynomial time even in certain games with chance moves. In triangle-free games, the algorithm of the authors can represent the space of correlation plans using a polynomial number of linear constraints with a quadratic number of variables. In the experimental evaluation, the authors show that the algorithm can construct the representations even for sequential games of larger size very fast.

Strengths: The authors have done a good job in presenting their theoretical results. All notions are well described, and examples accompany the most important concepts so the reader can quickly get familiar with them. I find the results significant and novel, and I believe they will give rise to new algorithms for computing correlated equilibria in sequential games. Overall, I really enjoyed this paper, and I think it makes a valuable contribution to the literature relevant to the NeurIPS community.

Weaknesses: I am aware that the paper's primary focus is the complexity refinement, and the empirical evaluation is not a central topic. Still, in the end, the decomposition should serve to find correlated equilibria more easily. My main (but minor) concerns are hence related to the application of the introduced decomposition. First, did the authors try to compare their algorithm to the algorithm of Dudík and Gordon (A Sampling-Based Approach to Computing Equilibria in Succinct Extensive-Form Games)? It would be interesting to see, e.g., the infeasibility over time for regret minimization vs. the algorithm of Dudík and Gordon. Second, you compute a feasible EFCE without an objective function, which is known to be a polynomial problem in any multi-player sequential game (by Huang, 2008) to start with, if I am not wrong. Therefore it does not take full advantage of your approach in this work. Could you explain why? Is it because you aim to compare regret minimization with linear programming, and the regret-minimization algorithm could not handle an objective? Did you perform any tests with EFCE with an objective function in triangle-free sequential games too? If yes, did you observe any drop in performance in terms of runtime when compared with EFCE without an objective function? Minor note: Could you make clear in Theorem 2 that s_i’s are simplex dimensions?

Correctness: I read through the proofs, and as far as I can tell, they seem correct. In the empirical evaluation, as expected, the authors chose sufficiently large triangle-free games with chance moves to show how their decomposition algorithm scales. Moreover, I am impressed that one-threaded regret minimization performs so well despite Gurobi running in 30 threads.

Clarity: The paper is very well written, and I did notice only a few typos (mentioned below). I read through all the proofs, and as far as I can tell, they seem to be correct. I find it impressive that despite sequential game theory being quite heavy on notation at times, the authors managed to explain their results without confusing the reader with too many symbols. Minor notes: 1/ In Equation 1, the first sum should iterate over A_{I_1}, not A_I and the second sum over A_{I_2}, not A_J. 2/ On line 244, you might be referring to a “leftmost” graph rather than a “topmost” graph. 3/ On line 566, set -> sets.

Relation to Prior Work: The paper describes well how the current research fits into the literature on the topic.

Reproducibility: Yes

Additional Feedback: ---- comments after feedback Thank you for answering my questions. I appreciate your comments on the last three points, regarding optimal correlated equilibria. I still have a few comments/nits, though: 1/ Please cite the work of Dudík and Gordon. It seems relevant to your work, given that literature on EFCE is still very scarce, even though their algorithm might be slow, numerically unstable, and is tailored for maximizing entropy. 2/ Huang gives an explicit LP for EFCE in Theorem 4 of [16] that enables to find optimal EFCE with a linear objective. It can indeed be used to compute optimal equilibria, but not in polynomial time.

[Author Response · NeurIPS 2020]

**Reviewer 1**: Thanks for the feedback and for the suggestions as to how to make the paper clearer and the examples less intimidating. We'll work on that for the final version. ● Re *"...how decomposing the polytope now allows it to be mapped?"* We are not sure we understand the question. If you meant "how does the decomposition help map the problem of computing an optimal correlated equilibrium onto an LP?", here is the answer. If the goal is to write an LP for computing an optimal correlated equilibrium in a triangle-free game, our crucial result is that $\Xi = \mathcal{V}$, and the scaled-extension decomposition can just be regarded as a detail in the proof of that equality. The LP for computing an optimal correlated equilibrium is of the form $\arg \max_{\boldsymbol{\xi}} \{ \boldsymbol{c}^\top \boldsymbol{\xi} : \boldsymbol{A\xi} + \boldsymbol{By} = \boldsymbol{0}, \boldsymbol{y} \geq \boldsymbol{0}, \boldsymbol{\xi} \in \Xi \}$, so once it is known that $\Xi = \mathcal{V}$, one can substitute the constraint $\boldsymbol{\xi} \in \Xi$ with $\boldsymbol{\xi} \in \mathcal{V}$, which corresponds to the (polynomially small) set of linear constraints given in Equation 1. ● Re *"I wasn't sure what I was supposed to take away from the experiments"* As mentioned, the scaled-extension-based decomposition of $\mathcal{V}$ is not directly used to write a linear program. However, it can be used to construct an efficient regret minimization algorithm for the set $\mathcal{V}$, which in turn can be used to compute an EFCE, EFCCE, NFCCE [12] (see also Reviewer 5). In the experiments, we show that (i) we implemented and tested the decomposition algorithm, and that it is able to scale to large games, and (ii) that we were able to experimentally confirm that the regret minimization algorithm is more scalable than the linear programming approach in large games—both in terms of run time and memory—(using the same experimental setup as [12]).

**Reviewer 2**: Thanks for the constructive feedback and suggestions as to how to improve the presentation and make the paper less intimidating. We'll take all of them into account. ● Re *"broader impact"* Thanks for the feedback, we agree with all your points. As you correctly recognized, we use the term "social welfare" to mean the sum of utilities of the players as is typical in the game theory literature, but as you rightly point out, that need not necessarily coincide with the societal notion of optimality. ● Re *"scale of infeasibility"* The constraints in Equation (1) are not the only constraints in the LP. Also, the sum of entries of a generic vector that satisfies the constraints in (1) will sum to more than 1 usually (i.e., correlation plans have more than unit mass). Gurobi's initial point when using its implementation of the barrier algorithm might be way off in terms of satisfying the constraint of the (primal) LP, and that is why Gurobi reported that the first iterate had a very high infeasibility (defined in lines 754-761). The maximum payoff is 15. ● Re *"free LP solver"* Thanks for the feedback. Gurobi is freely available for academic use, but we'll also mention the open-source (and slower, less numerically stable) alternative GLPK, unless the reviewers have better suggestions. ● Re *"Goofspiel"* We are not aware of it having been solved for EFCE before; we'll check. We are definitely the first to compute *optimal* EFCE in it. We will point these out as additional minor contributions of our paper. We will add results for all values of $k$ too, including plots with iteration count on the x-axis as suggested.

**Reviewer 3**: Thanks for your feedback. We strongly disagree that "*this paper just tells us that the work in Farina et al. [12] is straightforward to extend to the triangle-free game*" First we disagree that the extension is "straightforward"—and so seem to disagree the other reviewers too. Isolating and introducing triangle-freeness as a meaningful condition with consequences on the correlation structure of games is an important insight in itself. Extending the construction by Farina et al. to handle the more general structure is another (not easy) endeavor. Understanding how the existence of a decomposition relates to the integrality of the vertices of the von Stengel-Forges polytope is a third, separate result. On an independent note, we argue that whether the technical insights in a paper are simple or not in hindsight is not in itself a good metric for measuring the quality and potential impact of research. The other reviewers seem to think that the paper borders on too heavy technical contributions. You seem to suggest that it is too light on technical contribution. We strongly disagree with that. We hope the other reviews and our responses will convince you of the merits of the paper. ● Re *"line 244"*: we meant leftmost. We'll fix the typo. ● Re *"infeasibility"*. Thanks for the feedback. We'll make sure to define it in the body. Currently, a definition is available in the appendix, lines 754-761.

**Reviewer 5**: Thanks for pointing out typos and for the improvement suggestions! We'll apply them in the final version. ● Re *"the decomposition should serve to find correlated equilibria more easily"* The equality $\mathcal{V} = \Xi$ that we proved via the decomposition makes it *possible at all* (not just *easier*) to *optimize* over the set of correlated equilibria in polynomial time. This is because $\mathcal{V}$ can be expressed as the intersection of few linear constraints, while $\Xi$ in general cannot. (See also Reviewer 1.) ● Re *"does not take full advantage of your approach in this work"* While our focus is on offering the first algorithms to compute an optimal EFCE (in triangle-free games), we agree that it is likely that they would also be faster than the prior methods for finding a feasible equilibrium: 1) Dudik&Gordon requires convergences of MCMC at each iteration, likely making it slow and numerically unstable; 2) Huang is believed to be of theoretical interest rather than practically fast—because it uses the ellipsoid method. However, we leave investigating that direction as future work. In the last bit of the experimental section, we were simply interested in recreating the same setup as [12] to show that their conclusion hold in triangle-free games as well. Finally, we note that D&G and Huang cannot be used to compute optimal equilibria. ● Re *"the regret-minimization algorithm could not handle an objective?"* While we do not do that (and neither do the authors in [12]), the regret minimization method in [12] can find an equilibrium with a given lower bound on an objective. Hence, to optimize an objective one could, in theory, perform a binary search on the optimum objective value by running the regret minimization method several times with different lower bounds. However, that has never been tried in the literature. ● Re *"...any tests...EFCE with an objective function..."* We do use an objective to compute the sets in Figure 3 (right). In a nutshell, what we did was to take a game and try objective functions of the form $\alpha \cdot$ payoff player $1 + \beta \cdot$ payoff player 2 (with $\|(\alpha, \beta)\|_2 = 1$). Given a choice of $(\alpha, \beta)$, we computed the optimal objective value $v_{\alpha, \beta}$ of any EFCE, EFCCE, NFCCE of the game by solving the corresponding linear program with Gurobi. This shows that the set of reachable payoffs must satisfy the inequality $\alpha \cdot$ payoff player $1 + \beta \cdot$ payoff player $2 \leq v_{\alpha, \beta}$. Taking the intersection of all these constraints yields the polytope of reachable payoffs of the game. We'll include these and more details in the final version. ● Re *"did you observe any drop in performance...without an objective function?"* We did not extensively compare running the linear program with and without an objective on the same game instances, but our best guess informed from past experience with these equilibria is that the difference would be minimal when using Gurobi.

[Meta-Review · NeurIPS 2020]

The paper presents new results for computing extensive-form correlated equilibria in games with public chance actions. The author response clarified several points, and the main point of discussion was whether the content of this paper is new and significant enough in light of [12]. Though one reviewer clearly does not think so, the others do, and I lean toward agreeing that there is enough novelty in this paper to warrant its publication. However, there is an outstanding point that the authors have not responded to, which is how common these games are. At first glance this class of games does feel slightly artificial to me. The authors should state something on this, give some examples, or some discussion on what these games are or could be used for. I also encourage the authors to take all other points of feedback into account when revising the paper.